# Variance Based Sample Weighting for Supervised Deep Learning

## Abstract

In the context of supervised learning of a function by a Neural Network (NN), we claim and empirically justify that a NN yields better results when the distribution of the data set focuses on regions where the function to learn is steeper. We first traduce this assumption in a mathematically workable way using Taylor expansion. Then, theoretical derivations allow to construct a methodology that we call Variance Based Samples Weighting (VBSW). VBSW uses local variance of the labels to weight the training points. This methodology is general, scalable, cost effective, and significantly increases the performances of a large class of NNs for various classification and regression tasks on image, text and multivariate data. We highlight its benefits with experiments involving NNs from shallow linear NN to ResNet (He et al., 2015) or Bert (Devlin et al., 2019).

## 1 Introduction

When a Machine Learning (ML) model is used to learn from data, the distribution of the training data set can have a strong impact on its performances. More specifically, in the context of Deep Learning (DL), several works have hinted at the importance of the training set. In Bengio et al. (2009); Matiisen et al. (2017), the authors exploit the observation that a human will benefit more from easy examples than from harder ones at the beginning of a learning task. They construct a curriculum, inducing a change in the distribution of the training data set that makes a Neural Network (NN) achieve better results in an ML problem. With a different approach, Active Learning (Settles, 2012) modifies dynamically the distribution of the training data, by selecting the data points that will make the training more efficient. Finally, in Reinforcement Learning, the distribution of experiments is crucial for the agent to learn efficiently. Nonetheless, the challenge of finding a good distribution is not specific to ML. Indeed, in the context of Monte Carlo estimation of a quantity of interest based on a random variable $X \sim d\mathbb{P}_X$, Importance Sampling owes its efficiency to the construction of a second random variable, $\bar{X} \sim d\mathbb{P}_{\bar{X}}$ that will be used instead of $X$ to improve the estimation of this quantity. Jie & Abbeel (2010) even make a connection between the success of likelihood ratio policy gradients and importance sampling, which shows that ML and Monte Carlo estimation, both distribution based methods, are closely linked.

In this paper, we leverage the importance of the training set distribution to improve performances of NNs in supervised DL. This task can be formalized as approximating a function $f$ with a model $f_\theta$ parametrized by $\theta$. We build a new distribution from the training points and their labels, based on the observation that $f_\theta$ needs more data points to approximate $f$ on the regions where it is steep. We use Taylor expansion of a function $f$, which links the local behaviour of $f$ to its derivatives, to build this distribution. We show that up to a certain order and locally, variance is an estimator of Taylor expansion. It allows constructing a methodology called Variance Based Sample Weighting (VBSW) that weights each training data points using the local variance of their neighbor labels to simulate the new distribution. Sample weighting has already been explored in many works and for various goals. Kumar et al. (2010); Jiang et al. (2015) use it to prioritize easier samples for the training, Shrivastava et al. (2016) for hard example mining, Cui et al. (2019) to avoid class imbalance, or (Liu & Tao, 2016) to solve noisy label problem. In this work, the weights' construction relies on a more general claim that can be applied to any data set and whose goal is to improve the performances of the model.

VBSW is general, because it can be applied to any supervised ML problem based on a loss function. In this work we specifically investigate VBSW application to DL. In that case, VBSW is applied within the feature space of a pre-trained NN. We validate VBSW for DL by obtaining performance improvement on various tasks like classification and regression of text, from Glue benchmark (Wang et al., 2019), image, from MNIST (LeCun & Cortes, 2010) and Cifar10 (Krizhevsky et al.) and multivariate data, from UCI ML repository (Dua & Graff, 2017), for several models ranging from linear regression to Bert (Devlin et al., 2019) or ResNet20 (He et al., 2015). As a highlight, we obtain up to $1.65\%$ classification improvement on Cifar10 with a ResNet. Finally, we conduct analyses on the complementarity of VBSW with other weighting techniques and its robustness.

**Contributions**: **(i)** We present and investigate a new approach of the learning problem, based on the variations of the function $f$ to learn. **(ii)** We construct a new simple, scalable, versatile and cost effective methodology, VBSW, that exploits these findings in order to boost the performances of a NN. **(iii)** We validate VBSW on various ML tasks.

## 2 RELATED WORKS

**Active Learning** - Our methodology is based on the consideration that not every sample bring the same amount of information. Active learning (AL) exploits the same idea, in the sense that it adapts the training strategy to the problem by introducing a data point selection rule. In (Gal et al., 2017), the authors introduce a methodology based on Bayesian Neural Networks (BNN) to adapt the selection of points used for the training. Using the variational properties of BNN, they design a rule to focus the training on points that will reduce the prediction uncertainty of the NN. In (Konyushkova et al., 2017), the construction of the selection rule is taken as a ML problem itself. See (Settles, 2012) for a review of more classical AL methods. While AL selects the data points, so modifies the distribution of the initial training data set, VBSW is applied independently of the training so the distribution of the weights can not change throughout the training.

**Examples Weighting** - VBSW can be categorized as an example weighting algorithm. The idea of weighting the data set has already been explored in different ways and for different purposes. While curriculum learning (Bengio et al., 2009; Matiisen et al., 2017) starts the training with easier examples, Self paced learning (Kumar et al., 2010; Jiang et al., 2015) downscales harder examples. However, some works have proven that focusing on harder examples at the beginning of the learning could accelerate it. In (Shrivastava et al., 2016), hard example mining is performed to give more importance to harder examples by selecting them primarily. Example weighting is used in (Cui et al., 2019) to tackle the class imbalance problem by weighting rarer, so harder examples. At the contrary, in (Liu & Tao, 2016) it is used to solve the noisy label problem by focusing on cleaner, so easier examples. All these ideas show that depending on the application, example weighting can be performed in an opposed manner. Some works aim at going beyond this opposition by proposing more general methodologies. In (Chang et al., 2017), the authors use the variance of the prediction of each point throughout the training to decide whether it should be weighted or not. A meta learning approach is proposed in (Ren et al., 2018), where the authors choose the weights after an optimization loop included in the training. VBSW stands out from the previously mentioned example weighting methods because it is built on a more general assumption that a model would simply need more points to learn more complicated functions. Its effect is to improve the performances of a NN, without solving data set specific problems like class imbalance or noisy labels.

**Importance Sampling** - Some of the previously mentioned methods use importance sampling to design the weights of the data set or to correct the bias induced by the sample selection (Katharopoulos & Fleuret, 2018). Here, we construct a new distribution that could be interpreted as an importance distribution. However, we weight the data points to simulate this distribution, not to correct a bias induced by this distribution.

**Generalization Bound** - Generalization bound for the learning theory of NN have motivated many works, most of which are reviewed in (Jakubovitz et al., 2018). In Bartlett et al. (1998), Bartlett et al. (2019), the authors focus on VC-dimension, a measure which depends on the number of parameters of NNs. Arora et al. (2018) introduces a compression approach that aims at reducing the number of model parameters to investigate its generalization capacities. PAC-Bayes analysis constructs generalization bounds using *a priori* and *a posteriori* distributions over the possible models. It is

investigated for example in Neyshabur et al. (2018); Bartlett et al. (2017), and Neyshabur et al. (2017); Xu & Mannor (2012) links PAC-Bayes theory to the notion of sharpness of a NN, i.e. its robustness to small perturbation. While sharpness of the model is often mentioned in the previous works, our bound includes the derivatives of $f$, which can be seen as an indicator of the sharpness of the function to learn. Even if it uses elements of previous works, like the Lipschitz constant of $f_\theta$, our work does not pretend to tighten and improve the already existing generalization bounds, but only emphasizes the intuition that the NN would need more points to capture sharper functions. In a sense, it investigates the robustness to perturbations in the input space, not in the parameter space.

## 3 A New Training Distribution based on Taylor Expansion

In this section, we first illustrate why a NN may need more points where $f$ is steep by deriving a generalization bound that involves the derivatives of $f$. Then, using Taylor expansion, we build a new training distribution that improves the performances of a NN on simple functions.

### 3.1 Problem Formulation

We formalize the supervised ML task as approximating a function $f : \mathbf{S} \subset \mathbb{R}^{n_i} \to \mathbb{R}^{n_o}$ with an ML model $f_\theta$ parametrized by $\theta$, where $\mathbf{S}$ is a measured sub-space of $\mathbb{R}^{n_i}$ depending on the application. To this end, we are given a training data set of $N$ points, $\{X_1, ..., X_N\} \in \mathbf{S}$, drawn from $X \sim d\mathbb{P}_X$ and their point-wise values, or labels $\{f(X_1), ..., f(X_N)\}$. Parameters $\theta$ have to be found in order to minimize an integrated loss function $J_X(\theta) = \mathbb{E}_X[L(f_\theta(X), f(X))]$, with $L$ the loss function, $L : \mathbb{R}^{n_o} \times \mathbb{R}^{n_o} \to \mathbb{R}$. The data allow estimating $J_X(\theta)$ by $\widehat{J_X}(\theta) = \frac{1}{N} \sum_{i=1}^N L(f_\theta(X_i), f(X_i))$. Then, an optimization algorithm is used to find a minimum of $\widehat{J_X}(\theta)$ w.r.t. $\theta$.

### 3.2 Intuition behind Taylor Expansion

In the following, we illustrate the intuition with a Generalization Bound (GB) that include the derivatives of $f$, provided that these derivatives exist. The goal of the approximation problem is to be able to generalize to points not seen during the training. The generalization error $\mathcal{J}_X(\theta) = J_X(\theta) - \widehat{J_X}(\theta)$ thus needs to be as small as possible. Let $S_i, i \in \{1, ..., N\}$ be some sub-spaces of $\mathbf{S}$ such that $\mathbf{S} = \bigcup_{i=1}^N S_i$, $\bigcap_{i=1}^N S_i = \emptyset$, and $X_i \in S_i$. Suppose that $L$ is the squared $L_2$ error, $n_i = 1$, $f$ is differentiable and $f_\theta$ is $K_\theta$-Lipschitz. Provided that $|S_i| < 1$, we show that

$$\mathcal{J}_X(\theta) \leq \sum_{i=1}^N (|f'(X_i)| + K_\theta)^2 \frac{|S_i|^3}{4} + O(|S_i|^4), \tag{1}$$

where $|S_i|$ is the volume of $S_i$. The proof can be found in **Appendix B**. We see that on the regions where $f'(X_i)$ is higher, quantity $|S_i|$ has a stronger impact on the GB. This idea is illustrated on Figure 1. Since $|S_i|$ can be seen as a metric for the local density of the data set (the smaller $|S_i|$ is,

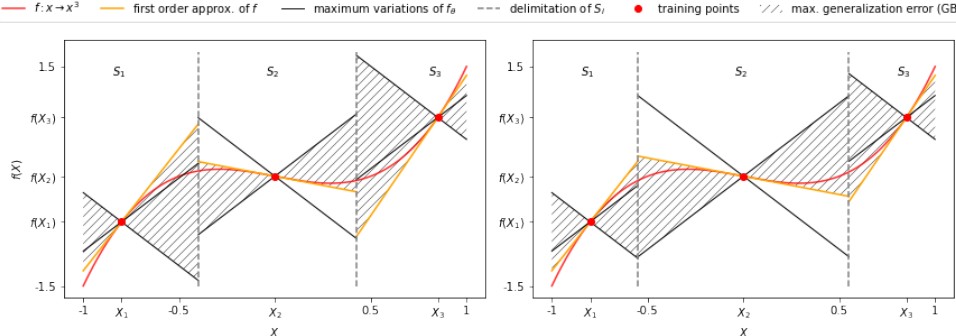

Figure 1: Illustration of the GB. The maximum error (the GB), at order $O(|S_i|^4)$, is obtained by comparing the maximum variations of $f_\theta$, and the first order approximation of $f$, whose trends are given by $K_\theta$ and $f'(X_i)$. We understand visually that because $f'(X_1)$ and $f'(X_3)$ are higher than $f'(X_2)$, the GB is improved more efficiently by reducing $S_1$ and $S_3$ than $S_2$.

the denser the data set is), the GB can be reduced more efficiently by adding more points around $X_i$ in these regions. This bound also involves $K_\theta$, the Lipschitz constant of the NN, which has the same impact than $f'(X_i)$. It also illustrates the link between the Lipschitz constant and the generalization error, which has been pointed out by several works like (Gouk et al., 2018), (Bartlett et al., 2017) and (Qian & Wegman, 2019). Note that equation 1 only gives indications about $n = 1$. Indeed, this GB only has illustration purposes. Its goal is to motivate the metric described in the next section, which is based on Taylor expansion and therefore involves derivatives of order $n > 1$.

### 3.3 A TAYLOR EXPANSION BASED METRIC

In this paragraph, we build a metric involving the derivatives of $f$. Using Taylor expansion at order $n$ on $f$ and supposing that $f$ is $n$ times differentiable (multi index notation):

$$f(x+\epsilon) \underset{\|\epsilon\|\to 0}{=} \sum_{0 \le |\boldsymbol{k}| \le n} \epsilon^{\boldsymbol{k}} \frac{\partial^{\boldsymbol{k}} f(x)}{\boldsymbol{k}!} + O(\epsilon^{\boldsymbol{n}}). \qquad Df_\epsilon^n(x) = \sum_{1 \le |\boldsymbol{k}| \le n} \frac{\|\epsilon\|^k \cdot \|\operatorname{Vect}(\partial^{\boldsymbol{k}} f(x))\|}{\boldsymbol{k}!}. \quad (2)$$

Quantity $f(x + \epsilon) - f(x)$ gives an indication on how much $f$ changes around $x$. By neglecting the orders above $\epsilon^{\boldsymbol{n}}$, it is then possible to find the regions of interest by focusing on $Df_\epsilon^n$, given by equation 2, where $\operatorname{Vect}(\mathbf{X})$ denotes the vectorization of a tensor $\mathbf{X}$ and $\|.\|$ the squared $L_2$ norm. Note that $Df_\epsilon^n$ is evaluated using $\|\partial^{\boldsymbol{k}} f(x)\|$ instead of $\partial^{\boldsymbol{k}} f(x)$ for derivatives not to cancel each other. $f$ will be steeper and more irregular in the regions where $x \to Df_\epsilon^n(x)$ is higher.

To focus the training set on these regions, one can use $\{Df_\epsilon^n(X_1), ..., Df_\epsilon^n(X_N)\}$ to construct a probability density function (pdf) and sample new data points from it. This sampling is evaluated and validated in **Appendix A** for conciseness. Based on these experiments, we choose $n = 2$, i.e. we use $\{Df_\epsilon^2(X_1), ..., Df_\epsilon^2(X_N)\}$. The good results obtained, presented in **Appendix A** confirm our observation and motivate its application to complex DL problems.

## 4 VARIANCE BASED SAMPLES WEIGHTING (VBSW)

### 4.1 PRELIMINARIES

The new distribution cannot always be applied as is, because we do not have access to $f$. **Problem 1:** $\{Df_\epsilon^2(X_1), ..., Df_\epsilon^2(X_N)\}$ cannot be evaluated since it requires to compute the derivatives of $f$. Moreover, it assumes that $f$ is differentiable, which is often not true. **Problem 2:** even if $\{Df_\epsilon^2(X_1), ..., Df_\epsilon^2(X_N)\}$ could be computed and new points sampled, we could not obtain their labels to complete the training data set.

**Problem 1: Unavailability of derivatives** To overcome **problem 1**, we construct a new metric based on statistical estimation. In this paragraph, $n_i > 1$ but $n_o = 1$. The following derivations can be extended to $n_o > 1$ by applying it to $f$ element-wise and then taking the sum across the $n_o$ dimensions. Let $\epsilon \sim \mathcal{N}(0, \epsilon \mathbf{I}_{n_i})$ with $\epsilon \in \mathbf{R}^+$ and $\mathbf{I}_{n_i}$ the identity matrix of dimension $n_i$. We claim that

$$Var(f(x + \epsilon)) = Df_\epsilon^2(x) + O(\|\epsilon\|_2^3).$$

The demonstration can be found in **Appendix B**. Using the unbiased estimator of variance, we thus define new indices $\widehat{Df_\epsilon^2}(x)$ by

$$\widehat{Df_\epsilon^2}(x) = \frac{1}{k-1} \sum_{i=1}^{k} \left( f(x + \epsilon_i) - f(x) \right)^2, \qquad (3)$$

with $\{\epsilon_1, ..., \epsilon_k\}$ $k$ samples of $\epsilon$. The metric $\widehat{Df^2}_\epsilon(x) \underset{k\to\infty}{\to} Var(f(x + \epsilon))$ and $Var(f(x + \epsilon)) = Df_\epsilon^2(x) + O(\|\epsilon\|_2^3)$, so $\widehat{Df^2}_\epsilon(x)$ is a biased estimator of $Df_\epsilon^2(x)$, with bias $O(\|\epsilon\|_2^3)$. Hence, when $\epsilon \to 0$, $\widehat{Df^2}_\epsilon(x)$ becomes an unbiased estimator of $Df_\epsilon^2(x)$. It is possible to compute $\widehat{Df^2}(x)$ from any set of points centered around $x$. Therefore, we compute $\widehat{Df^2}(X_i)$ for each $i \in \{1, ..., N\}$ using the set $\mathcal{S}_k(X)$ of $k$-nearest neighbors of $X_i$. We obtain $\widehat{Df^2}(X_i)$ using

$$\widehat{Df^2}(X_i) = \frac{1}{k-1} \sum_{X_j \in \mathcal{S}_k(X_i)} \left( f(X_j) - \frac{1}{k} \sum_{l=1}^{k} f(X_l) \right)^2, \tag{4}$$

The advantages of this formulation are twofold. First, $\widehat{Df^2}$ can even be applied to non-differentiable functions. Second, all we need is $\{f(X_1), ..., f(X_N)\}$. In other words, the points used by $\widehat{Df^2}(X_i)$ are those used for the training of the NN. Finally, while the definition of $Df_\epsilon^2(x)$ is local, the definition of $\widehat{Df^2}_\epsilon(x)$ holds for any $\epsilon$. Note that equation 4 can even be applied when the data points are too sparse for the nearest neighbors of $X_i$ to be considered as close to $X_i$. It can thus be seen as a generalization of $Df_\epsilon^2(x)$, which tends towards $Df_\epsilon^2(x)$ locally.

**Problem 2: Unavailability of new labels** To tackle **problem 2**, recall that the goal of the training is to find $\theta^* = \underset{\theta}{\operatorname{argmin}} \widehat{J_X}(\theta)$, with $\widehat{J_X}(\theta) = \frac{1}{N} \sum_i L(f(X_i), f_\theta(X_i))$. With the new distribution based on previous derivations, the procedure is different. Since the training points are sampled using $\widehat{Df^2}_\epsilon$, we no longer minimize $\widehat{J_X}(\theta)$, but $\widehat{J_{\bar{X}}}(\theta) = \frac{1}{N} \sum_i L(f(\bar{X}_i), f_\theta(\bar{X}_i))$, with $\bar{X} \sim d\mathbb{P}_{\bar{X}}$ the new distribution. However, $\widehat{J_{\bar{X}}}(\theta)$ estimates

$$J_{\bar{X}}(\theta) = \int_{\mathbf{S}} L(f(x), f_\theta(x)) d\mathbb{P}_{\bar{X}}.$$

Let $p_X(x)dx = d\mathbb{P}_X$, $p_{\bar{X}}(x)dx = d\mathbb{P}_{\bar{X}}$ be the pdfs of $X$ and $\bar{X}$ (note that $Df_\epsilon^2 \propto p_{\bar{X}}$). Then,

$$J_{\bar{X}}(\theta) = \int_{\mathbf{S}} L(f(x), f_\theta(x)) \frac{p_{\bar{X}}(x)}{p_X(x)} d\mathbb{P}_X.$$

The straightforward Monte Carlo estimator for this expression of $J_{\bar{X}}(\theta)$ is

$$\widehat{J_{\bar{X},2}}(\theta) = \frac{1}{N} \sum_i L(f(X_i), f_\theta(X_i)) \frac{p_{\bar{X}}(X_i)}{p_X(X_i)} \propto \frac{1}{N} \sum_i L(f(X_i), f_\theta(X_i)) \frac{\widehat{Df^2}(X_i)}{p_X(X_i)}. \tag{5}$$

Thus, $J_{\bar{X}}(\theta)$ can be estimated with the same points as $J_X(\theta)$ by weighting them with $w_i = \frac{\widehat{Df^2}(X_i)}{p_X(X_i)}$.

## 4.2 Hyperparameters of VBSW

The expression of $w_i$ involves $Df_\epsilon^2(X_i)$, whose estimation has been the goal of the previous sections. However, it also involves $p_X$, the distribution of the data. Just like for $f$, we do not have access to $p_X$. The estimation of $p_X$ is a challenging task by itself, and standard density estimation techniques such as K-nearest neighbors or Gaussian Mixture density estimation led to extreme estimated values of $p_X(X_i)$ in our experiments. Therefore, we decided to only apply $\omega_i = \widehat{Df^2}(X_i)$ as a first order approximation. In practice, we re-scale the weighting points to be between 1 and $m$, a hyperparameter. As a result, VBSW has two hyperparameters: $m$ and $k$. Their effects and interactions are studied and discussed in Sections 5.1 and 5.4.

## 4.3 VBSW for Deep Learning

We specified that local variance could be computed using already existing points. This statement implies to find the nearest neighbors of each point. In extremely high dimension spaces like image spaces the curse of dimensionality makes nearest neighbors spurious. In addition, the structure of the data may be highly irregular, and the concept of nearest neighbor misleading. Thus, it may be irrelevant to evaluate $\widehat{D^2 f_\epsilon}$ directly on this data.

One of the strength of DL is to construct good representations of the data, embedded in lower dimensional latent spaces. For instance, in Computer Vision, Convolutional Neural Networks (CNN)'s deeper layers represent more abstract features. We could leverage this representational power of NNs, and simply apply our methodology within this latent feature space.

---

**Algorithm 1** Variance Based Samples Weighting (VBSW) for Deep learning

1: **Inputs:** $k, m, \mathcal{M}$
2: Train $\mathcal{M}$ on the training set $\{(\frac{1}{N}, X_1), ..., (\frac{1}{N}, X_N)\}, \{(\frac{1}{N}, f(X_1)), ..., (\frac{1}{N}, f(X_N))\}$
3: Construct $\mathcal{M}^*$ by removing its last layer
4: Compute $\{\widehat{Df^2}(\mathcal{M}^*(X_1)), ..., \widehat{Df^2}(\mathcal{M}^*(X_N))\}$ using equation 4.
5: Construct a new training data set $\{(w_1, \mathcal{M}^*(X_1)), ..., (w_N, \mathcal{M}^*(X_N))\}$
6: Train $f_\theta$ on $\{(w_1, f(X_1)), ..., (w_N, f(X_N))\}$ and add it to $\mathcal{M}^*$. The final model is $\mathcal{M}_f = f_\theta \circ \mathcal{M}^*$

---

Variance Based Samples Weighting (VBSW) for DL is recapitulated in Algorithm 1. Here, $\mathcal{M}$ is the initial NN whose feature space will be used to project the training data set and apply VBSW. **Line 1:** $m$ and $k$ are hyperparameters that can be chosen jointly with all other hyperparameters, e.g. using a random search. **Line 2:** The initial NN, $\mathcal{M}$, is trained as usual. Notations $\{(\frac{1}{N}, X_1), ..., (\frac{1}{N}, X_N)\}$ is equivalent to $\{X_1, ..., X_N\}$, because all the weights are the same ($\frac{1}{N}$). **Line 3:** The last fully connected layer is discarded, resulting in a new model $\mathcal{M}^*$, and the training data set is projected in the feature space. **Line 4-5:** equation 4 is applied to compute the weights $w_i$ that are used to weight the projected data set. To perform nearest neighbors search, we use KD-Tree (Bentley, 1975). **Line 6:** The last layer is re-trained (which is often equivalent to fitting a linear model) using the weighted data set and added to $\mathcal{M}^*$ to obtain the final model $\mathcal{M}_f$. As a result, $\mathcal{M}_f$ is a composition of the already trained model $\mathcal{M}^*$ and $f_\theta$ trained using the weighted data set.

## 5 EXPERIMENTS

We first test this methodology on toy datasets with linear models and small NNs. Then, to illustrate how general VBSW can be, we consider various tasks in image classification, text regression and classification. Finally, we study the robustness of VBSW and its complementarity with other sample weighting techniques.

### 5.1 TOY EXPERIMENTS

VBSW is studied on a Double Moon (DM) classification, in the Boston Housing (BH) regression and Breast Cancer (BC) classification data sets.

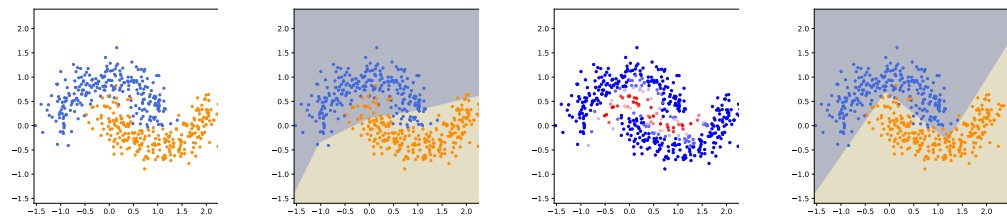

Figure 2: From left to right: **(a)** Double Moon (DM) data set. **(b)** Decision boundary with the baseline method. **(c)** Heat map of the value of $w_i$ for each $X_i$ (red is high and blue is low) and **(d)** Decision boundary with VBSW method

For DM, Figure 2 (c) shows that the points with highest $w_i$ (in red) are close to the boundary between the two classes. Indeed, in classification, VBSW can be interpreted as local label agreement. We train a Multi Layer Perceptron of 1 layer of 4 units, using Stochastic Gradient Descent (SGD) and binary cross-entropy loss function, on a 300 points training data set for

|     | VBSW | baseline |
|-----|------|----------|
| DM | **99.4**, **94.44** $\pm 0.78$ | $99, 92.06 \pm 0.66$ |
| BH | **13.31**, **13.38** $\pm 0.01$ | $14.05, 14.06 \pm 0.01$ |
| BC | **99.12**, $97.6 \pm 0.34$ | $98.25, 97.5 \pm 0.11$ |

Table 1: **best, mean + se** for each method. The metric used is accuracy for DM and BC and Mean Squared Error for BH.

50 random seeds. In this experiment, VBSW, i.e. weighting the data set with $w_i$ is compared to baseline where no weights are applied. Figure 2 (b) and (d) displays the decision boundary of best

fit for each method. VBSW provides a cleaner decision boundary than baseline. These pictures as well as the results of Table 1 show the improvement obtained with VBSW.

For BH data set, a linear model is trained and for BC data set, a MLP of 1 layer and 30 units, with a train-validation split of $80\% - 20\%$. Both models are trained with ADAM (Kingma & Ba, 2014). Since these data sets are small and the models are light, we study the effects of the choice of $m$ and $k$ on the performances. Moreover, BH is a regression task and BC a classification task, so it allows studying the effect of hyperparameters more extensively. We train the models for a grid of $20 \times 20$ different values of $m$ and $k$. These hyperparameters seem to have a different impact on performances for classification and regression. In both cases, low values for $m$ yields better results, but in classification, low values of $k$ are better, unlike in regression. Details and visualization of this experiment can be found in **Appendix C**. The best results obtained with this study are compared to the best result of the same models trained without VBSW in Table 1.

## 5.2 MNIST AND CIFAR10

For MNIST, we train 40 LeNet 5, i.e. with 40 different random seeds, and then apply VBSW for 10 different random seeds, with ADAM optimizer and categorical cross-entropy loss. Note that in the following, ADAM is used with the default parameters of its `keras` implementation. We record the best value obtained from the 10 VBSW training. The same procedure is followed for Cifar10, except that we train a ResNet20 for 50 random seeds and with data augmentation and learning rate decay. The networks have been trained on 4 Nvidia K80 GPUs. The values of the hyperparameters used can be found in **Appendix C**. We compare the test accuracy between LeNet 5 + VBSW, ResNet20 + VBSW and the initial test accuracies of LeNet 5 and ResNet20 (baseline) for each of the initial networks.

|         | VBSW                        | baseline                 | gain per model           |
|---------|-----------------------------|--------------------------|--------------------------|
| MNIST   | **99.09**, **98.87** $\pm 0.01$ | $98.99, 98.84 \pm 0.01$  | **0.15**, **0.03** $\pm 0.01$ |
| Cifar10 | **91.30**, **90.64** $\pm 0.07$ | $91.01, 90.46 \pm 0.10$  | **1.65**, **0.15** $\pm 0.04$ |

Table 2: **best, mean + se** for each method. The metric used is accuracy. For a model $\mathcal{M}$, the gain $g$ for this model is given by $g = \max_{1 \le i \le 10} (acc(\mathcal{M}_f^i) - acc(\mathcal{M}))$ with $acc$ the accuracy and $\mathcal{M}_f^i$ the VBSW model trained at the $i$-th random seed.

The results statistics are gathered in Table 2, which also displays statistics about the gain due to VBSW for each model. The results on MNIST, for all statistics and for the gain are significantly better than forVBSW than for baseline. For Cifar10, we get a $0.3\%$ accuracy improvement for the best model and up to $1.65\%$ accuracy gain, meaning that among the 50 ResNet20s, there is one whose accuracy has been improved by $1.65\%$ by VBSW. Note that applying VBSW took less than 15 minutes on a laptop with an i7-7700HQ CPU. A visualization of the samples that were weighted by the highest $w_i$ is given in Figure 3.

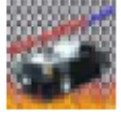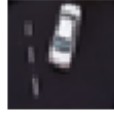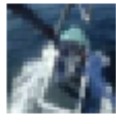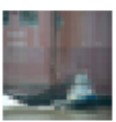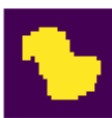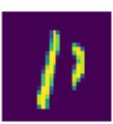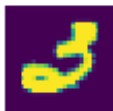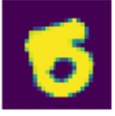

Figure 3: Samples from Cifar10 and MNIST with high $w_i$. Those pictures are either unusual or difficult to classify, even for a human (especially for MNIST).

## 5.3 RTE, STS-B AND MRPC

For this application, we do not pre-train Bert NN, like in the previous experiments, since it has been originally built for Transfer Learning purposes. Therefore, its purpose is to be used as is and then fine-tuned on any NLP data set see (Devlin et al., 2019). However, because of the small size of the dataset and the high number of model parameters we chose not to fine-tune the Bert model, and only to use the representations of the data sets in its feature space to apply VBSW. More specifically, we use tiny-bert (Turc et al., 2019), which is a lighter version of the initial Bert NN. We train the linear model with tensorflow, to be able to add the trained model on top of the Bert model and obtain a

unified model. RTE and MRPC are classification tasks, so we use binary cross-entropy loss function to train our model. STS-B is a regression task so the model is trained with Mean Squared Error. All the models are trained with ADAM optimizer. For each task, we compare the training of the linear model with VBSW, and without VBSW (baseline). The results obtained with VBSW are better overall, except for Pearson Correlation in STS-B, which is slightly worse than baseline (Table 3).

| | VBSW | | baseline | |
| | m1 | m2 | m1 | m2 |
|---|---|---|---|---|
| RTE | **61.73**, **58.46** $\pm 0.15$ | - | $61.01, 58.09 \pm 0.13$ | - |
| STS-B | **62.31**, **62.20** $\pm 0.01$ | **60.99**, $60.88 \pm 0.01$ | $61.88, 61.87 \pm 0.01$ | $60.98$, **60.92** $\pm 0.01$ |
| MRPC | **72.30**, **71.71** $\pm 0.03$ | **82.64**, **80.72** $\pm 0.05$ | $71.56, 70.92 \pm 0.03$ | $81.41, 80.02 \pm 0.07$ |

Table 3: **best, mean + se** for each method. For RTE the metric used is accuracy (m1). For MRPC, metric 1 (m1) is accuracy and metric 2 (m2) is F1 score. For STS-B, metric 1 (m1) is Spearman correlation and metric 2 (m2) is Pearson correlation.

## 5.4 ROBUSTNESS OF VBSW

VBSW relies on statistical estimation: the weights are based on local empirical variance, evaluated using $k$ points. In addition, they are rescaled using hyperparameter $m$. Section 5.1 and **Appendix C** show that many different combinations of $m$ and $k$ and therefore many different values for the weights improve the error. This behavior suggests that VBSW is quite robust to weights approximation error.

We also assess the robustness of VBSW to label noise. To that end, we train a ResNet20 on Cifar10 with four different noise levels. We randomly change the label of $p\%$ training points for four different values of $p$ (10, 20, 30 and 40). We then apply VBSW 30 times and evaluate the performances of the obtained NNs on a clean test set. The results are gathered in Table 4.

| noise | 10% | 20% | 30% | 40% |
|---|---|---|---|---|
| original error | 87.43 | 85.75 | 84.05 | 81.79 |
| VBSW | **87.76**, **87.63** $\pm 0.01$ | **86.03**, **85.89** $\pm 0.01$ | **84.35**, **84.18** $\pm 0.02$ | **82.48**, **82.32** $\pm 0.02$ |

Table 4: **best, mean + se** of the training of a ResNet20 on Cifar10 for different label noise levels. These results illustrate the robustness of VBSW to labels noise.

The results show that VBSW is still effective despite label noise, which could be explained by two observations. First, the weights of VBSW rely on statistical estimation, so perturbations in the labels might have a limited impact on weights' value. Second, as mentioned previously, VBSW is robust to weights approximation error, so perturbation of the weights due to label noise may not critically hurt the method. Although VBSW is robust to label noise, note that the goal of VBSW is not to address noisy label problem, like discussed in Section 2. It may be more effective to use a sampling technique specifically tailored for this situation - possibly jointly with VBSW, like in Section 5.5.

## 5.5 COMPLEMENTARITY OF VBSW

Existing sample weighting techniques can be used jointly with VBSW by training the initial NN $\mathcal{M}$ with the first sample weighting algorithm, and then applying VBSW on its feature space. To illustrate this feature, we compare VBSW with the recently introduced Active Bias (AB) (Chang et al., 2017). AB dynamically weights the samples based on the variance of the probability of prediction of each points throughout the training. Here, we study the combined effects of AB and VBSW for the training of a ResNet20 on Cifar10. Table 5 gathers the results of experiments for different baselines: vanilla, for a regular training with Adam optimizer, AB for a training with AB, VBSW for the application of VBSW on top of a regular training and VBSW + AB for a training with AB and the application of VBSW. Unlike in section 5.2, we do not use data augmentation or learning rate decay, to simplify the experiments (which explains the accuracy loss compared to previous experiments).

These results demonstrate the competitiveness of VBSW compared with AB and their complementarity. Indeed, the accuracy obtained with VBSW is quite similar to AB and the best mean and max accuracy is obtained for a NN trained with AB + VBSW. Note that in this experiment, the gain

|  | accuracy (%) | VBSW gain per model |
|---|---|---|
| vanilla | $75.88, 74.55 \pm 0.11$ | - |
| AB | $76.33, 75.14 \pm 0.09$ | - |
| VBSW | $76.57, 74.94 \pm 0.10$ | $\mathbf{0.94}, \mathbf{0.40} \pm 0.03$ |
| AB + VBSW | $\mathbf{76.60}, \mathbf{75.33} \pm \mathbf{0.09}$ | $0.40, 0.014 \pm 0.02$ |

Table 5: **best, mean + se** of the training of 60 ResNet20s on Cifar10 for vanilla, VBSW, AB and AB + VBSW. Gain per model $g$ is defined by $g = \max_{1 \leq i \leq 10} (acc(\mathcal{M}_f^i) - acc(\mathcal{M}))$ with $acc$ the accuracy and $\mathcal{M}_f^i$ the VBSW model trained at the $i$-th random seed.

per model is lower for AB + VBSW than for VBSW alone. An explanation might be that AB is already improving the NN performances compared to vanilla, so there is less room for accuracy improvement by VBSW in that case.

## 6 DISCUSSION & FUTURE WORK

Previous experiments demonstrate the performances improvement that VBSW can bring in practice. In addition to these results, several advantages can be pointed out.

- VBSW is validated on several different tasks, which makes it quite versatile. Moreover, the problem of high dimensionality and irregularity of $f$, which often arises in DL problems, is alleviated by focusing on the latent space of NNs. This makes VBSW scalable. As a result, VBSW can be applied to complex NNs such as ResNet, a complex CNN or Bert, for various ML tasks. Its application to more diverse ML models is a perspective for future works.

- The validation presented in this paper supports an original view of the learning problem, that involves the local variations of $f$. The studies of **Appendix A**, that use the derivatives of the function to learn to sample a more efficient training data set, support this approach as well.

- VBSW allows to extend this original view to problems where the derivatives of $f$ are not accessible, and sometimes not defined. Indeed, VBSW comes from Taylor expansion, which is specific to derivable functions, but in the end can be applied regardless of the properties of $f$.

- Finally, this method is cost effective. In most cases, it allows to quickly improve the performances of a NN using a regular CPU. In terms of energy consumption, it is better than carrying on a whole new training with a wider and deeper NN.

We first approximated $p_X$ to be uniform, because we could not approximate it correctly. This approximation still led to a an efficient methodology, but VBSW may benefit from a finer approximation of $p_X$. Improving the approximation of $p_X$ is among our perspectives. Finally, the KD-tree and even Approximate Nearest Neighbors algorithms struggle when the data set is too big. One possibility to overcome this problem would be to parallelize their execution.

We only considered the cases where we have no access to $f$. However, there are ML applications where we do. For instance, in numerical simulations, for physical sciences, computational economics or climatology, ML can be used for various reasons, e.g. sensitivity analysis, inverse problems or to speed up computer codes (Zhu et al., 2019; Winovich et al., 2019; Feng et al., 2018). In this context data comes from numerical models, so the derivatives of $f$ are accessible and could be directly used. **Appendix A** contains examples of such possible applications.

## 7 CONCLUSION

Our work is based on the observation that, in supervised learning, a function $f$ is more difficult to approximate by a NN in the regions where it is steeper. We mathematically traduced this intuition and derived a generalization bound to illustrate it. Then, we constructed an original method, Variance Based Samples Weighting (VBSW), that uses the variance of the training samples to weight the training data set and boosts the model's performances. VBSW is simple to use and to implement, because it only requires to compute statistics on the input space. In Deep Learning, applying VBSW on the data set projected in the feature space of an already trained NN allows to reduce its error by simply training its last layer. Although specifically investigated in Deep Learning, this method is applicable to any loss function based supervised learning problem, scalable, cost effective, robust and versatile. It is validated on several applications such as glue benchmark with Bert, for text classification and regression and Cifar10 with a ResNet20, for image classification.

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

## APPENDIX A TAYLOR BASED SAMPLING

In this part, we empirically verify that using Taylor expansion to construct a new training distribution has a beneficial impact on the performances of a NN. To this end, we construct a methodology, that we call Taylor Based Sampling (TBS), that generates a new training data set based on the metric introduced in Section **3.3**. First, we recall the formula for $\{Df_{\boldsymbol{\epsilon}}^n(X_1), ..., Df_{\boldsymbol{\epsilon}}^n(X_N)\}$.

$$Df_{\boldsymbol{\epsilon}}^n(x) = \sum_{1 \leq |\boldsymbol{k}| \leq n} \frac{\|\boldsymbol{\epsilon}\|^k \cdot \|\operatorname{Vect}(\partial^{\boldsymbol{k}} f(x))\|}{\boldsymbol{k}!}. \tag{6}$$

To focus the training set on the regions of interest, i.e. regions of high $\{Df_{\boldsymbol{\epsilon}}^n(X_1), ..., Df_{\boldsymbol{\epsilon}}^n(X_N)\}$, we use this metric to construct a probability density function (pdf). This is possible since $Df_{\boldsymbol{\epsilon}}^n(x) \geq 0$ for all $x \in \mathbf{S}$. It remains to normalize it but in practice it is enough considering a distribution $d \propto Df_{\boldsymbol{\epsilon}}^n$. Here, to approximate $d$ we use a Gaussian Mixture Model (GMM) with pdf $d_{\text{GMM}}$ that we fit to $\{Df_{\boldsymbol{\epsilon}}^n(X_1), ..., Df_{\boldsymbol{\epsilon}}^n(X_N)\}$ using the Expectation-Maximization (EM) algorithm. $N'$ new data points $\{\bar{X}_1, ..., \bar{X}_{N'}\}$, can be sampled, with $\bar{X} \sim d_{\text{GMM}}$. Finally, we obtain $\{f(\bar{X}_1), ..., f(\bar{X}_{N'})\}$, add it to $\{f(X_1), ..., f(X_N)\}$ and train our NN on the whole data set.

### TAYLOR BASED SAMPLING (TBS)

TBS is described in Algorithm 2. **Line 1:** The choice criterion of $\epsilon$, the number of Gaussian distribution $n_{\text{GMM}}$ and $N'$ is to avoid sparsity of $\{\bar{X}_1, ..., \bar{X}_{N'}\}$ over $\mathbf{S}$. **Line 2:** Without *a priori* information on $f$, we sample the first points uniformly in a subspace $\mathbf{S}$. **Line 3-7:** We construct $\{Df_{\boldsymbol{\epsilon}}^n(X_1), ..., Df_{\boldsymbol{\epsilon}}^n(X_N)\}$, and then $d$ to be able to sample points accordingly. **Line 8:** Because the support of a GMM is not bounded, some points can be sampled outside $\mathbf{S}$. We discard these points and sample until all points are inside $\mathbf{S}$. This rejection method is equivalent to sampling points from a truncated GMM. **Line 9-10:** We construct the labels and add the new points to the initial data set.

---

**Algorithm 2** Taylor Based Sampling (TBS)

---

1: **Inputs:** $\epsilon$, $N$, $N'$, $n_{\text{GMM}}$, $n$
2: Sample $\{X_1, ..., X_N\}$, with $X \sim \mathcal{U}(\mathbf{S})$
3: **for** $0 \leq k \leq n$ **do**
4:     Compute $\{\partial^{\boldsymbol{k}} f(X_1), ..., \partial^{\boldsymbol{k}} f(X_N)\}$
5: Compute $\{Df_{\boldsymbol{\epsilon}}^n(X_1), ..., Df_{\boldsymbol{\epsilon}}^n(X_N)\}$ using equation 2
6: Approximate $d \sim D_{\boldsymbol{\epsilon}}$ with a GMM using EM algorithm to obtain a density $d_{\text{GMM}}$
7: Sample $\{\bar{X}_1, ..., \bar{X}_{N'}\}$ using rejection method to sample inside $\mathbf{S}$
8: Compute $\{f(\bar{X}_1), ..., f(\bar{X}_{N'})\}$
9: Add $\{f(\bar{X}_1), ..., f(\bar{X}_{N'})\}$ to $\{f(X_1), ..., f(X_N)\}$

---

### APPLICATION TO SIMPLE FUNCTIONS

In order to illustrate the benefits of TBS compared to a uniform, basic sampling (BS), we apply it to two simple functions: hyperbolic tangent and Runge function. We chose these functions because they are differentiable and have a clear distinction between flat and steep regions. These functions are displayed in Figure 4, as well as the map $x \to Df_{\boldsymbol{\epsilon}}^2(x)$.

All NN have been implemented in `Python`, with `Tensorflow` Abadi et al. (2015). We use the Python package `scikit-learn` Pedregosa et al. (2011), and more specifically the `GaussianMixture` class to construct

| Sampling | $L_2$ error | $L_\infty$ |
|---|---|---|
| | $f$: Runge ($\times 10^{-2}$) | |
| BS | $1.45 \pm 0.62$ | $5.31 \pm 0.86$ |
| **TBS** | $\mathbf{1.13 \pm 0.73}$ | $\mathbf{3.87 \pm 0.48}$ |
| | $f$: tanh ($\times 10^{-1}$) | |
| BS | $1.39 \pm 0.67$ | $2.75 \pm 0.78$ |
| **TBS** | $\mathbf{0.95 \pm 0.50}$ | $\mathbf{2.25 \pm 0.61}$ |

Table 6: Comparison between BS and TBS. The metrics used are the $L_2$ and $L_\infty$ errors, displayed with a $95\%$ confidence interval.

$d_{GMM}$. The network chosen for this experiment is a Multi Layer Perceptron (MLP) with one layer of 8 neurons, with Relu activation function, that we trained alternatively with BS and TBS using

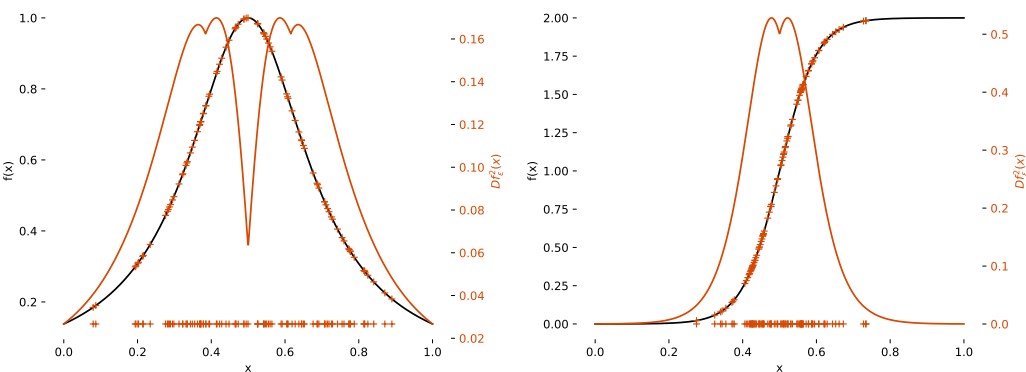

Figure 4: **Left:** (left axis) Runge function w.r.t x and (right axis) $x \rightarrow Df_{\epsilon}^2(x)$. Points sampled using TBS are plotted on the x-axis and projected on $f$. **Right:** Same as left, with hyperbolic tangent function.

Adam optimizer Kingma & Ba (2014) with the defaults tensorflow implementation hyperparameters, and Mean Squared Error loss function. We first sample $\{X_1, ..., X_N\}$ according to a regular grid. To compare the two methods, we add $N'$ additional points sampled using BS to create the BS data set, and then $N'$ other points sampled with TBS to construct the TBS data set. As a result, each data set have the same number of points $(N + N')$. We repeated the method for several values of $n$, $n_{\text{GMM}}$ and $\epsilon$, and finally selected $n = 2$, $n_{\text{GMM}} = 3$ and $\epsilon = 10^{-3}$.

Table 6 summarizes the $L_2$ and the $L_{\infty}$ norm of the error of $f_{\theta}$, obtained at the end of the training phase for $N + N' = 16$, with $N = 8$. Those norms are estimated using a same test data set of 1000 points. The values are the means of the 40 independent experiments displayed with a 95% confidence interval. These results illustrate the benefits of TBS over BS. Table 6 shows that TBS slightly degrades the $L_2$ error of the NN, but improves its $L_{\infty}$ error. This may explain the good results of VBSW for classification. Indeed, for a classification task, the accuracy will not be very sensitive to small output variations, since the output is rounded to 0 or 1. However, a high error can induce a misclassification, and the reduction in $L_{\infty}$ norm limits this risk.

APPLICATION TO AN ODE SYSTEM

We apply TBS to a more realistic case: the approximation of the resolution of the Bateman equations, which is an ODE system :

$$\begin{cases} \partial_t u(t) & = v\boldsymbol{\sigma_a} \cdot \boldsymbol{\eta}(t)u(t), \\ \partial_t \boldsymbol{\eta}(t) & = v\boldsymbol{\Sigma_r} \cdot \boldsymbol{\eta}(t)u(t), \end{cases} \text{, with initial conditions } \begin{cases} u(0) = u_0, \\ \boldsymbol{\eta}(0) = \boldsymbol{\eta_0}. \end{cases},$$

with $u \in \mathbb{R}^+, \boldsymbol{\eta} \in (\mathbb{R}^+)^M, \boldsymbol{\sigma}_a^T \in \mathbb{R}^M, \boldsymbol{\Sigma}_r \in \mathbb{R}^{M \times M}$. Here, $f : (u_0, \boldsymbol{\eta_0}, t) \rightarrow (u(t), \boldsymbol{\eta}(t))$. For physical applications, $M$ ranges from tens to thousands. We consider the particular case $M = 1$ so that $f : \mathbb{R}^3 \rightarrow \mathbb{R}^2$, with $f(u_0, \eta_0, t) = (u(t), \eta(t))$. The advantage of $M = 1$ is that we have access to an analytic, cheap to compute solution for $f$. Of course, this particular case can also be solved using a classical ODE solver, which allows us to test it end to end. It can thus be generalized to higher dimensions $(M > 1)$.

All NN trainings have been performed in `Python`, with `Tensorflow` Abadi et al. (2015). We used a fully connected NN with hyperparameters chosen using a simple grid search. The final values are: 2 hidden layers, "ReLU" activation function, and 32 units for each layer, trained with the Mean Squared Error (MSE) loss function using Adam optimization algorithm with a batch size of 50000, for 40000 epochs and on $N + N' = 50000$ points, with $N = N'$. We first trained the model for $(u(t), \eta(t)) \in \mathbb{R}$, with an uniform sampling (BS) $(N' = 0)$, and then with TBS for several values of $n$, $n_{\text{GMM}}$ and $\epsilon = \epsilon(1, 1, 1)$, to be able to find good values. We finally select $\epsilon = 5 \times 10^{-4}$, $n = 2$ and $n_{\text{GMM}} = 10$. The data points used in this case have been sampled with an explicit Euler

scheme. This experiment has been repeated 50 times to ensure statistical significance of the results.

**Table 7** summarizes the MSE, i.e. the $L_2$ norm of the error of $f_\theta$ and $L_\infty$ norm, with $L_\infty(\theta) = \max_{X \in \mathbf{S}}(|f(X) - f_\theta(X)|)$ obtained at the end of the training phase. This last metric is important because the goal in computational physics is not only to be averagely accurate, which is measured with MSE, but to be accurate over the whole input space $\mathbf{S}$. Those norms are estimated using a same test data set of $N_{test} = 50000$ points. The values are the means of the 50 independent experiments displayed with a 95% confidence interval. These results reflect an error reduction of 6.6% for $L_2$ and of 45.3% for $L_\infty$, which means that TBS mostly improves the $L_\infty$ error of $f_\theta$. Moreover, the $L_\infty$ error confidence intervals do not intersect so the gain is statistically significant for this norm.

Table 7: Comparison between BS and TBS

| Sampling | $L_2$ error ($\times 10^{-4}$) | $L_\infty$ ($\times 10^{-1}$) | AEG($\times 10^{-2}$) | AEL($\times 10^{-2}$) |
|---|---|---|---|---|
| BS | $1.22 \pm 0.13$ | $5.28 \pm 0.47$ | - | - |
| **TBS** | $\mathbf{1.14 \pm 0.15}$ | $\mathbf{2.96 \pm 0.37}$ | $2.51 \pm 0.07$ | $0.42 \pm 0.008$ |

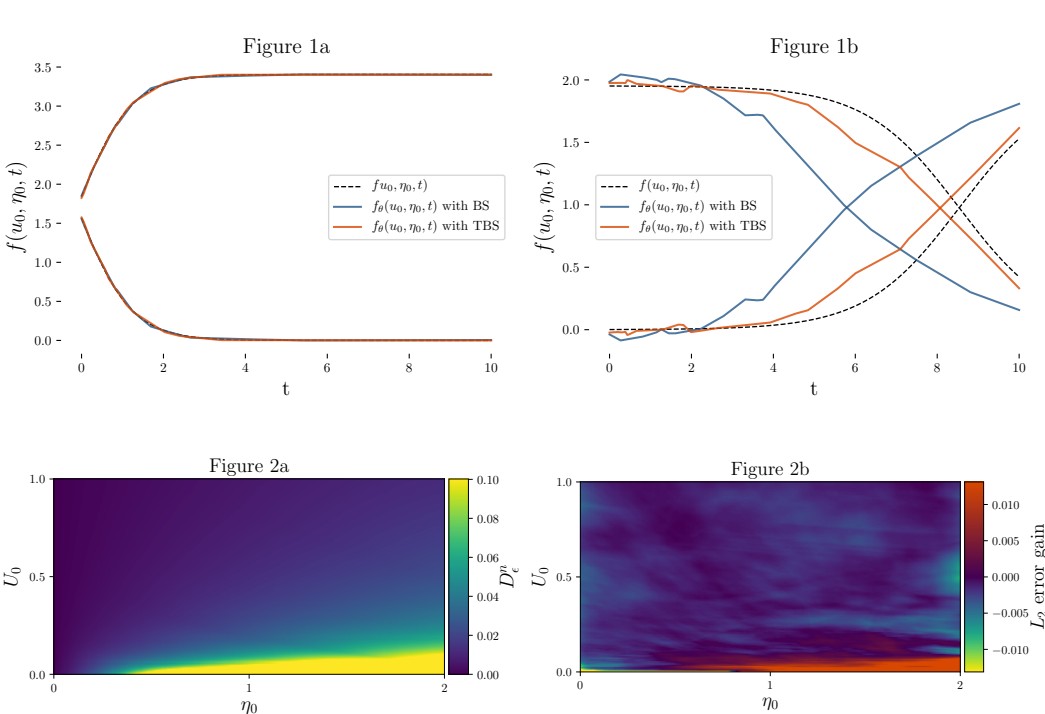

Figure 5: **1a:** $t \to f_\theta(u_0, \eta_0, t)$ for randomly chosen $(u_0, \eta_0)$, for $f_\theta$ obtained with the two samplings. **1b:** $t \to f_\theta(u_0, \eta_0, t)$ for $(u_0, \eta_0)$ resulting in the highest point-wise error with the two samplings. **2a:** $u_0, \eta_0 \to \max_{0 \le t \le 10} D_\epsilon^n(u_0, \eta_0, t)$ w.r.t. $(u_0, \eta_0)$. **2b:** $u_0, \eta_0 \to g_{\theta_{BS}}(u_0, \eta_0) - g_{\theta_{TBS}}(u_0, \eta_0)$,

**Figure 1a** shows how the NN can perform for an average prediction. **Figure 1b** illustrates the benefits of TBS relative to BS on the $L_\infty$ error (Figure 2b). These 2 figures confirm the previous observation about the gain in $L_\infty$ error. Finally, **Figure 2a** displays $u_0, \eta_0 \to \max_{0 \le t \le 10} D_\epsilon^n(u_0, \eta_0, t)$ w.r.t. $(u_0, \eta_0)$ and shows that $D_\epsilon^n$ increases when $U_0 \to 0$. TBS hence focuses on this region. Note that for the readability of these plots, the values are capped to 0.10. Otherwise only few points with high $D_\epsilon^n$ are visible. **Figure 2b** displays $u_0, \eta_0 \to g_{\theta_{BS}}(u_0, \eta_0) - g_{\theta_{TBS}}(u_0, \eta_0)$, with $g_\theta : u_0, \eta_0 \to \max_{0 \le t \le 10} \|f(u_0, \eta_0, t) - f_\theta(u_0, \eta_0, t)\|_2^2$ where $\theta_{BS}$ and $\theta_{TBS}$ denote the parameters obtained after a training with BS and TBS, respectively. It can be interpreted as the error reduction achieved with TBS.

The highest error reduction occurs in the expected region. Indeed, more points are sampled where $D_\epsilon^n$ is higher. The error is slightly increased in the rest of **S**, which could be explained by a sparser sampling on this region. However, as summarized in **Table 1**, the average error loss (AEL) of TBS is around six times lower than the average error gain (AEG), with $AEG = \mathbb{E}_{u_0,\eta_0}(Z\mathbf{1}_{Z>0})$ and $AEL = \mathbb{E}_{u_0,\eta_0}(Z\mathbf{1}_{Z<0})$ where $Z(u_0,\eta_0) = g_{\theta_{BS}}(u_0,\eta_0) - g_{\theta_{TBS}}(u_0,\eta_0)$. In practice, AEG and AEL are estimated using uniform grid integration, and averaged on the 50 experiments.

## APPENDIX B: DEMONSTRATIONS

### INTUITION BEHIND TAYLOR EXPANSION (SECTION 3.2)

We want to approximate $f : x \to f(x)$, $x \in \mathbb{R}^{n_i}$, $f(x) \in \mathbb{R}^{n_o}$ with a NN $f_\theta$. The goal of the approximation problem can be seen as being able to generalize to points not seen during the training. We thus want the generalization error $\mathcal{J}_X(\theta)$ to be as small as possible. Given an initial data set $\{X_1, ..., X_N\}$ drawn from $X \sim d\mathbb{P}_X$ and $\{f(X_1), ..., f(X_N)\}$, and the loss function $L$ being the squared $L_2$ error, recall that the integrated error $J_X(\theta)$, its estimation $\widehat{J_X}(\theta)$ and the generalization error $\mathcal{J}_X(\theta)$ can be written:

$$
\begin{aligned}
J_X(\theta) &= \int_{\mathbf{S}} \|f(x) - f_\theta(x)\| d\mathbb{P}_X, \\
\widehat{J_X}(\theta) &= \frac{1}{N}\sum_{i=1}^{N} \|f_\theta(X_i) - f(X_i)\|, \\
\mathcal{J}_X(\theta) &= J_X(\theta) - \widehat{J_X}(\theta),
\end{aligned}
\tag{7}
$$

where $\|.\|$ denotes the squared $L_2$ norm. In the following, we find an upper bound for $\mathcal{J}_X(\theta)$. We start by finding an upper bound for $J_X(\theta)$ and then for $\mathcal{J}_X(\theta)$ using equation 7.

Let $S_i$, $i \in \{1, ..., N\}$ be some sub-spaces of a bounded space **S** such that $\mathbf{S} = \bigcup_{i=1}^{N} S_i$, $\bigcap_{i=1}^{N} S_i = \emptyset$, and $X_i \in S_i$. Then,

$$
\begin{aligned}
J_X(\theta) &= \sum_{i=1}^{N} \int_{S_i} \|f(x) - f_\theta(x)\| d\mathbb{P}_X, \\
J_X(\theta) &= \sum_{i=1}^{N} \int_{S_i} \|f(X_i + x - X_i) - f_\theta(x)\| d\mathbb{P}_X.
\end{aligned}
$$

Suppose that $n_i = n_o = 1$ and $f$ twice differentiable. Let $|\mathbf{S}| = \int_{\mathbf{S}} d\mathbb{P}_X$. The volume $|\mathbf{S}| = 1$ since $d\mathbb{P}_X$ is a probability measure, and therefore $|S_i| < 1$ for all $i \in \{1, ..., N\}$. Using Taylor expansion at order 2, and since $|S_i| < 1$ for all $i \in \{1, ..., N\}$

$$
J_X(\theta) = \sum_{i=1}^{N} \int_{S_i} \|f(X_i) + f'(X_i)(x - X_i) + \frac{1}{2}f''(X_i)(x - X_i)^2 - f_\theta(x) + O((x - X_i)^3)\| d\mathbb{P}_X.
$$

To find an upper bound for $J(\theta)$, we can first find an upper bound for $|A_i(x)|$, with $A_i(x) = f(X_i) + f'(X_i)(x - X_i) + \frac{1}{2}f''(X_i)(x - X_i)^2 - f_\theta(x) + O((x - X_i)^3)$.

NN $f_\theta$ is $K_\theta$−Lipschitz, so since **S** is bounded (so are $S_i$), for all $x \in S_i$, $|f_\theta(x) - f_\theta(X_i)| \leq K_\theta |x - X_i|$. Hence,

$$f_\theta(X_i) - K_\theta |x - X_i| \le f_\theta(x) \le f_\theta(X_i) + K_\theta |x - X_i|,$$
$$- f_\theta(X_i) - K_\theta |x - X_i| \le -f_\theta(x) \le -f_\theta(X_i) + K_\theta |x - X_i|,$$
$$f(X_i) + f'(X_i)(x - X_i) + \frac{1}{2} f''(X_i(x - X_i)^2) - f_\theta(X_i) - K_\theta |x - X_i| + O((x - X_i)^3)$$
$$\le A_i(x) \le f(X_i) + f'(X_i)(x - X_i) + \frac{1}{2} f''(X_i)(x - X_i)^2 - f_\theta(X_i) + K_\theta |x - X_i| + O((x - X_i)^3),$$
$$A_i(x) \le f(X_i) - f_\theta(X_i) + f'(X_i)(x - X_i) + \frac{1}{2} f''(X_i)(x - X_i)^2 + K_\theta |x - X_i| + O((x - X_i)^3).$$

And finally, using triangular inequality,

$$\boxed{A_i(x) \le |f(X_i) - f_\theta(X_i)| + |f'(X_i)||x - X_i| + \frac{1}{2} |f''(X_i)||x - X_i|^2 + K_\theta |x - X_i| + O(|x - X_i|^3).}$$

Now, $\|.\|$ being the squared $L_2$ norm:

$$J_X(\theta) = \sum_{i=1}^{N} \int_{S_i} \|f(X_i) + f'(X_i)(x - X_i) + \frac{1}{2} f''(X_i)(x - X_i)^2 - f_\theta(x) + O(|x - X_i|^3)\| d\mathbb{P}_X,$$

$$J_X(\theta) \le \sum_{i=1}^{N} \int_{S_i} \left[ \left( |f(X_i) - f_\theta(X_i)| \right) + \left( |f'(X_i)||x - X_i| + \frac{1}{2} |f''(X_i)||x - X_i|^2 + K_\theta |x - X_i| \right) \right.$$
$$\left. + O(|x - X_i|^3) \right]^2 d\mathbb{P}_X,$$

$$= \sum_{i=1}^{N} \int_{S_i} \left[ |f(X_i) - f_\theta(X_i)|^2 \right.$$
$$+ 2|f(X_i) - f_\theta(X_i)| \left( |f'(X_i)||x - X_i| + \frac{1}{2} |f''(X_i)||x - X_i|^2 + K_\theta |x - X_i| \right)$$
$$\left. + \left[ \left( |f'(X_i)||x - X_i| \right) + \left( \frac{1}{2} |f''(X_i)||x - X_i|^2 + K_\theta |x - X_i| \right) \right]^2 + O(|x - X_i|^3) \right] d\mathbb{P}_X,$$

$$= \sum_{i=1}^{N} \int_{S_i} \left[ |f(X_i) - f_\theta(X_i)|^2 \right.$$
$$+ 2|f(X_i) - f_\theta(X_i)| \left( |f'(X_i)||x - X_i| + \frac{1}{2} |f''(X_i)||x - X_i|^2 + K_\theta |x - X_i| \right)$$
$$\left. + \left[ |f'(X_i)|^2 |x - X_i|^2 + 2K_\theta |f'(X_i)||x - X_i|^2 + K_\theta^2 |x - X_i|^2 \right] + O(|x - X_i|^3) \right] d\mathbb{P}_X,$$

$$= \sum_{i=1}^{N} \int_{S_i} \left[ |f(X_i) - f_\theta(X_i)|^2 \right.$$
$$+ 2|f(X_i) - f_\theta(X_i)| \left( |f'(X_i)||x - X_i| + \frac{1}{2} |f''(X_i)||x - X_i|^2 + K_\theta |x - X_i| \right)$$
$$\left. + \left( |f'(X_i)| + K_\theta \right)^2 |x - X_i|^2 + O(|x - X_i|^3) \right] d\mathbb{P}_X.$$

Hornik's theorem (Hornik et al., 1989) states that given a norm $\|.\|_{p,\mu} =$ such that $\|f\|_{p,\mu}^p = \int_S |f(x)|^p d\mu(x)$, with $d\mu$ a probability measure, for any $\epsilon$, there exists $\theta$ such that for a Multi Layer Perceptron, $f_\theta$, $\|f(x) - f_\theta(x)\|_{p,\mu}^p < \epsilon$,

This theorem grants that for any $\epsilon$, with $d\mu = \sum_{i=1}^{N} \frac{1}{N}\delta(x - X_i)$, there exists $\theta$ such that

$$\begin{cases} \|f(x) - f_\theta(x)\|_{1,\mu}^1 = \sum_{i=1}^{N} \frac{1}{N}|f(X_i) - f_\theta(X_i)| \leq \epsilon, \\ \|f(x) - f_\theta(x)\|_{2,\mu}^2 = \sum_{i=1}^{N} \frac{1}{N}\big(f(X_i) - f_\theta(X_i)\big)^2 \leq \epsilon. \end{cases} \tag{8}$$

Let's introduce $i^*$ such that $i^* = \arg\min |S_i|$. Note that for any $i \in \{1, ..., N\}$, $O(|S_i^*|^4)$ is $O(|S_i|^4)$. Now, let's choose $\epsilon$ such that $\epsilon = O(|S_i^*|^4)$. Then, equation 8 implies that

$$\begin{cases} |f(X_i) - f_\theta(X_i)| = O(|S_i|^4), \\ \big(f(X_i) - f_\theta(X_i)\big)^2 = O(|S_i|^4), \\ \widehat{J_X}(\theta) = \|f(x) - f_\theta(x)\|_{2,\mu}^2 = O(|S_i|^4). \end{cases}$$

Thus, we have $\mathcal{J}_X(\theta) = J_X(\theta) - \widehat{J_X}(\theta) = J_X(\theta) + O(|S_i|^4)$ and therefore,

$$\mathcal{J}_X(\theta) \leq \sum_{i=1}^{N} \int_{S_i} \left[\Big(|f'(X_i)| + K_\theta\Big)^2 |x - X_i|^2 d\mathbb{P}_X\right] + O(|S_i|^4).$$

Finally,

$$\boxed{\mathcal{J}_X(\theta) \leq \sum_{i=1}^{N}(|f'(X_i)| + K_\theta)^2 \frac{|S_i|^3}{3} + O(|S_i|^4).} \tag{9}$$

We see that on the regions where $f'(X_i) + K_\theta$ is higher, quantity $|S_i|$ (the volume of $S_i$) has a stronger impact on the GB. Then, since $|S_i|$ can be seen as a metric for the local density of the data set (the smaller $|S_i|$ is, the denser the data set is), the Generalization Bound (GB) can be reduced more efficiently by adding more points around $X_i$ in these regions. This bound also involves $K_\theta$, the Lipschitz constant of the NN, which has the same impact as $f'(X_i)$. It also illustrates the link between the Lipschitz constant and the generalization error, which has been pointed out by several works like, for instance, Gouk et al. (2018), Bartlett et al. (2017) and Qian & Wegman (2019).

PROBLEM 1: UNAVAILABILITY OF DERIVATIVES (SECTION 4.1)

In this paragraph, we consider $n_i > 1$ but $n_o = 1$. The following derivations can be extended to $n_o > 1$ by applying it to $f$ element-wise. Let $\epsilon \sim \mathcal{N}(0, \epsilon \mathbf{1}_{n_i})$ with $\epsilon \in \mathbb{R}^+$ and $\epsilon = (\epsilon_1, ..., \epsilon_{n_i})$, i.e. $\epsilon_i \sim \mathcal{N}(0, \epsilon)$. Using Taylor expansion on $f$ at order 2 gives

$$f(x + \epsilon) = f(x) + \nabla_x f(x) \cdot \epsilon + \frac{1}{2}\epsilon^T \cdot \mathbb{H}_x f(x) \cdot \epsilon + O(\|\epsilon\|_2^3).$$

With $\nabla_x f$ and $\mathbb{H}_x f(x)$ the gradient and the Hessian of $f$ w.r.t. $x$. We now compute $Var(f(X + \epsilon))$ and make $Df_\epsilon^2(x) = \epsilon\|\nabla_x f(x)\|_F^2 + \frac{1}{2}\epsilon^2\|\mathbb{H}_x f(x)\|_F^2$ appear in its expression to establish a link between these two quantities:

$$Var(f(x + \epsilon)) = Var\Big(f(x) + \nabla_x f(x) \cdot \epsilon + \frac{1}{2}\epsilon^T \cdot \mathbb{H}_x f(x) \cdot \epsilon + O(\|\epsilon\|_2^3)\Big),$$

$$= Var\Big(\nabla_x f(x) \cdot \epsilon + \frac{1}{2}\epsilon^T \cdot \mathbb{H}_x f(x) \cdot \epsilon\Big) + O(\|\epsilon\|_2^3).$$

Since $\epsilon_i \sim \mathcal{N}(0, \epsilon)$, $x = (x_1, ..., x_{n_i})$ and with $\frac{\partial^2 f}{\partial x_i x_j}(X)$ the cross derivatives of $f$ w.r.t. $x_i$ and $x_j$,

$$\nabla_x f(x) \cdot \epsilon + \frac{1}{2}\epsilon^T \cdot \mathbb{H}_x f(x) \cdot \epsilon = \sum_{i=1}^{n_i} \epsilon_i \frac{\partial f}{\partial x_i}(x) + \frac{1}{2}\sum_{j=1}^{n_i}\sum_{k=1}^{n_i} \epsilon_j \epsilon_k \frac{\partial^2 f}{\partial x_j x_k}(x),$$

$$
Var\Big(\nabla_x f(x) \cdot \epsilon + \frac{1}{2}\epsilon^T \cdot \mathbb{H}_x f(x) \cdot \epsilon\Big) = Var\Big(\sum_{i=1}^{n_i} \epsilon_i \frac{\partial f}{\partial x_i}(x) + \frac{1}{2}\sum_{j=1}^{n_i}\sum_{k=1}^{n_i} \epsilon_j \epsilon_k \frac{\partial^2 f}{\partial x_j x_k}(x)\Big),
$$

$$
= \sum_{i_1=1}^{n_i}\sum_{i_2=1}^{n_i} Cov\Big(\epsilon_{i_1}\frac{\partial f}{\partial x_{i_1}}(x), \epsilon_{i_2}\frac{\partial f}{\partial x_{i_2}}(x)\Big),
$$

$$
+ \frac{1}{4}\sum_{j_1=1}^{n_i}\sum_{k_1=1}^{n_i}\sum_{j_2=1}^{n_i}\sum_{k_2=1}^{n_i} Cov\Big(\epsilon_{j_1}\epsilon_{k_1}\frac{\partial^2 f}{\partial x_{j_1} x_{k_1}}(x), \epsilon_{j_2}\epsilon_{k_2}\frac{\partial^2 f}{\partial x_{j_2} x_{k_2}}(x)\Big)
$$

$$
+ \sum_{i=1}^{n_i}\sum_{j=1}^{n_i}\sum_{k=1}^{n_i} Cov\Big(\epsilon_i\frac{\partial f}{\partial x_i}(x), \epsilon_j\epsilon_k\frac{\partial^2 f}{\partial x_j x_k}(x)\Big),
$$

$$
= \sum_{i_1=1}^{n_i}\sum_{i_2=1}^{n_i} \frac{\partial f}{\partial x_{i_1}}(x)\frac{\partial f}{\partial x_{i_2}}(x) Cov\Big(\epsilon_{i_1}, \epsilon_{i_2}\Big)
$$

$$
+ \frac{1}{4}\sum_{j_1=1}^{n_i}\sum_{k_1=1}^{n_i}\sum_{j_2=1}^{n_i}\sum_{k_2=1}^{n_i} \frac{\partial^2 f}{\partial x_{j_1} x_{k_1}}(x)\frac{\partial^2 f}{\partial x_{j_2} x_{k_2}}(x) Cov\Big(\epsilon_{j_1}\epsilon_{k_1}, \epsilon_{j_2}\epsilon_{k_2}\Big)
$$

$$
+ \sum_{i=1}^{n_i}\sum_{j=1}^{n_i}\sum_{k=1}^{n_i} \frac{\partial f}{\partial x_i}(x)\frac{\partial^2 f}{\partial x_j x_k}(x) Cov\Big(\epsilon_i, \epsilon_j\epsilon_k\Big).
$$

In this expression, we have to assess three quantities: $Cov(\epsilon_{i_1}, \epsilon_{i_2})$, $Cov(\epsilon_i, \epsilon_j\epsilon_k)$ and $Cov(\epsilon_{j_1}\epsilon_{k_1}, \epsilon_{j_2}\epsilon_{k_2})$.

First, since $(\epsilon_1, ..., \epsilon_{n_i})$ are i.i.d.,

$$
Cov\Big(\epsilon_{i_1}, \epsilon_{i_2}\Big) = \begin{cases} Var(\epsilon_i) = \epsilon \text{ if } i_1 = i_2 = i, \\ 0 \text{ otherwise.} \end{cases}
$$

To assess $Cov(\epsilon_i, \epsilon_j\epsilon_k)$, three cases have to be considered.

- If $i = j = k$, because $\mathbb{E}[\epsilon_i^3] = 0$,

$$
\begin{aligned}
Cov(\epsilon_i, \epsilon_j\epsilon_k) &= Cov(\epsilon_i, \epsilon_i^2), \\
&= \mathbb{E}[\epsilon_i^3] - \mathbb{E}[\epsilon_i]\mathbb{E}[\epsilon_i^2], \\
&= 0.
\end{aligned}
$$

- If $i = j$ or $i = k$ (we consider $i = k$, and the result holds for $i = j$ by commutativity),

$$
\begin{aligned}
Cov(\epsilon_i, \epsilon_j\epsilon_k) &= Cov(\epsilon_i, \epsilon_i\epsilon_j), \\
&= \mathbb{E}[\epsilon_i^2\epsilon_j] - \mathbb{E}[\epsilon_i]\mathbb{E}[\epsilon_i\epsilon_j], \\
&= \mathbb{E}[\epsilon_i^2]\mathbb{E}[\epsilon_j], \\
&= 0.
\end{aligned}
$$

- If $i \neq j$ and $i \neq k$, $\epsilon_i$ and $\epsilon_j\epsilon_k$ are independent and so $Cov(\epsilon_i, \epsilon_j\epsilon_k) = 0$.

Finally, to assess $Cov(\epsilon_{j_1}\epsilon_{k_1}, \epsilon_{j_2}\epsilon_{k_2})$, four cases have to be considered:

- If $j_1 = j_2 = k_1 = k_2 = i$,

$$
\begin{aligned}
Cov(\epsilon_{j_1}\epsilon_{k_1}, \epsilon_{j_2}\epsilon_{k_2}) &= Var(\epsilon_i^2), \\
&= 2\epsilon^2.
\end{aligned}
$$

- If $j_1 = k_1 = i$ and $j_2 = k_2 = j$, $Cov(\epsilon_{j_1}\epsilon_{k_1}, \epsilon_{j_2}\epsilon_{k_2}) = Cov(\epsilon_i^2, \epsilon_j^2) = 0$ since $\epsilon_i^2$ and $\epsilon_j^2$ are independent.

- If $j_1 = j_2 = j$ and $k_1 = k_2 = k$,

$$Cov(\epsilon_{j_1}\epsilon_{k_1}, \epsilon_{j_2}\epsilon_{k_2}) = Var(\epsilon_j\epsilon_k),$$
$$= Var(\epsilon_j)Var(\epsilon_k),$$
$$= \epsilon^2.$$

- If $j_1 \neq k_1, j_2$ and $k_2$,

$$Cov(\epsilon_{j_1}\epsilon_{k_1}, \epsilon_{j_2}\epsilon_{k_2}) = \mathbb{E}[\epsilon_{j_1}\epsilon_{k_1}\epsilon_{j_2}\epsilon_{k_2}] - \mathbb{E}[\epsilon_{j_1}\epsilon_{k_1}]\mathbb{E}[\epsilon_{j_2}\epsilon_{k_2}],$$
$$= \mathbb{E}[\epsilon_{j_1}]\mathbb{E}[\epsilon_{k_1}\epsilon_{j_2}\epsilon_{k_2}] - \mathbb{E}[\epsilon_{j_1}]\mathbb{E}[\epsilon_{k_1}]\mathbb{E}[\epsilon_{j_2}\epsilon_{k_2}],$$
$$= 0.$$

All other possible cases can be assessed using the previous results, commutativity and symmetry of $Cov$ operator. Hence,

$$
\begin{aligned}
Var\left(\nabla_x f(x) \cdot \epsilon + \frac{1}{2}\epsilon^T \cdot \mathbb{H}_x f(x) \cdot \epsilon\right) &= \sum_{i_1=1}^{n_i}\sum_{i_2=1}^{n_i} \frac{\partial f}{\partial x_{i_1}}(x)\frac{\partial f}{\partial x_{i_2}}(x)Cov\left(\epsilon_{i_1}, \epsilon_{i_2}\right) \\
&\quad + \frac{1}{4}\sum_{j_1=1}^{n_i}\sum_{k_1=1}^{n_i}\sum_{j_2=1}^{n_i}\sum_{k_2=1}^{n_i} \frac{\partial^2 f}{\partial x_{j_1}x_{k_1}}(x)\frac{\partial^2 f}{\partial x_{j_2}x_{k_2}}(x)Cov\left(\epsilon_{j_1}\epsilon_{k_1}, \epsilon_{j_2}\epsilon_{k_2}\right), \\
&= \sum_{i=1}^{n_i} \epsilon\frac{\partial f^2}{\partial x_i}(x) + \frac{1}{2}\sum_{j=1}^{n_i}\sum_{k=1}^{n_i} \epsilon^2 \frac{\partial^2 f^2}{\partial x_j x_k}(x), \\
&= \epsilon\|\nabla_x f(x)\|_F^2 + \frac{1}{2}\epsilon^2\|\mathbb{H}_x f(x)\|_F^2, \\
&= Df_\epsilon^2(x).
\end{aligned}
$$

And finally,

$$\boxed{Var(f(x+\epsilon)) = Df_\epsilon^2(x) + O(\|\epsilon\|_2^3)} \tag{10}$$

If we consider $\widehat{Df^2}_\epsilon(x)$ as defined in equation 2, on section 3.3 of the main document, $\widehat{Df^2}_\epsilon(x) \underset{k\to\infty}{\to} Var(f(x+\epsilon))$. Since $Var(f(x+\epsilon)) = Df_\epsilon^2(x) + O(\|\epsilon\|_2^3)$, $\widehat{Df^2}_\epsilon(x)$ is a biased estimator of $Df_\epsilon^2(x)$, with bias $O(\|\epsilon\|_2^3)$. Hence, when $\epsilon \to 0$, $\widehat{Df^2}_\epsilon(x)$ becomes an unbiased estimator of $Df_\epsilon^2(x)$.

APPENDIX C: HYPERPARAMETERS

This appendix Section is split in two parts. First, we describe the results of the experiments on the hyperparameters search of Boston Housing (BH) and Breast Cancer (BC) data sets (Section 5.1). The second part is a list of final hyperparameters values chosen for the experiments of the main paper.

EXPERIMENTS ON BOSTON HOUSING AND BREAST CANCER DATA SETS

For BH and BC experiments, we conduct a grid search for VBSW on the values of $m$ and $k$. As a reminder, $m$ is the ratio between the highest and the lowest weights, and $k$ is the number of neighbor points used to compute the local variance. We train a linear model for BH and a MLP with 30 units for BC with VBSW on a grid of 20 values of $m$ equally distributed between 2 and 100 and 20 values of $k$ equally distributed between 10 and 50. As a result, we train the model on 400 pairs of $(m, k)$ values, and with 10 different random seeds for each pair.

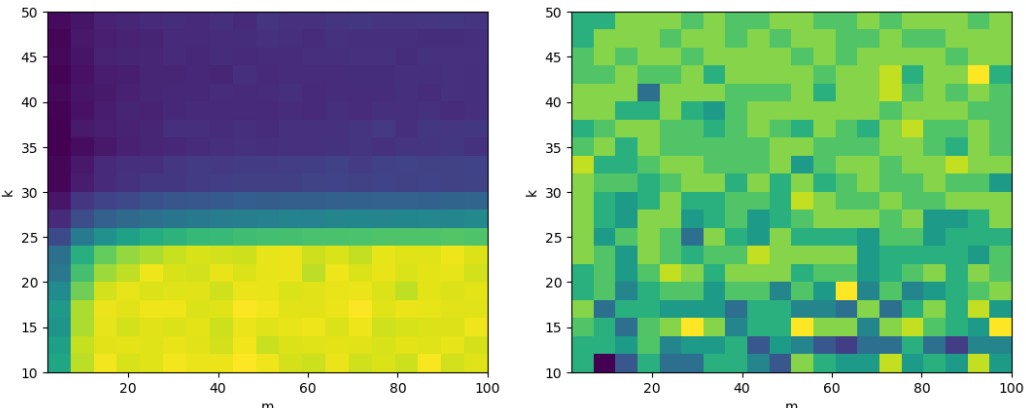

Figure 6: Color map of the error, with respect to $m$ and $k$. **Left:** BH data set, for the mean of the MSE accross 10 different seeds and **right:** BC data set, for the mean of $1 - acc$ across these seeds. Blue is lower.

These experiments, illustrated in Figure 6 shows that the influence of $m$ and $k$ on the performances of the model can be different. For BH data set, low values of $k$ clearly lead to poorer performances. Hyperparameter $m$ seems to have less impact, although it should be chosen not too far from its lowest value, 2. For BC data set, at the contrary, the best performances are obtained for low values of $k$, while $m$ could be chosen in high values. These experiments highlight that the impact of $m$ and $k$ can be different between classification and regression, but it could also be different depending on the data set. Hence, we recommend considering these hyperparameters like many other involved in DL, and to select their values using classical hyperparameters optimization techniques.

This also shows that many different $(m, k)$ pairs lead to error improvement. This suggests that the weights approximation does not have to be exact in order for VBSW to be effective, like stated in Section 5.4.

PAPER HYPERPARAMETERS VALUES

The values chosen for the hyperparameters of the paper experiments are gathered in Table 8. For ADAM optimizer hyperparameters, we kept the default values of Keras implementation. We chose these hyperparameters after simple grid searches.

| Experiment | $m$ | $k$ | learning rate | batch size | epochs | optimizer | random seeds |
|---|---|---|---|---|---|---|---|
| double moon | 100 | 20 | $1 \times 10^{-3}$ | 100 | 10000 | SGD | 50 |
| Boston housing | 8 | 35 | $5 \times 10^{-4}$ | 404 | 50000 | ADAM | 10 |
| Breast Cancer | 50 | 35 | $5 \times 10^{-2}$ | 455 | 250000 | ADAM | 10 |
| MNIST | 40 | 20 | $1 \times 10^{-3}$ | 25 | 25 | ADAM | 40 |
| Cifar10 | 40 | 20 | $1 \times 10^{-3}$ | 25 | 25 | ADAM | 50 |
| RTE | 20 | 10 | $3 \times 10^{-4}$ | 8 | 10000 | ADAM | 50 |
| STS-B | 30 | 30 | $3 \times 10^{-4}$ | 8 | 10000 | ADAM | 50 |
| MRPC | 75 | 25 | $3 \times 10^{-4}$ | 16 | 10000 | ADAM | 50 |

Table 8: Paper experiments hyperparameters values

