# OpenReview forum: "Variance Based Sample Weighting for Supervised Deep Learning"
_ICLR.cc/2021/Conference — Reject_

### Official Review · AnonReviewer2 · 2020-10-26
**In this study, a strategy called VBSW is proposed to simulate a new distribution over the training set by assigning weights to the data points using their local variance. This strategy is used as a preprocessing stage, which serves to prepare the data for NN learners.  Formally, Taylor expansion is used to describe the local behavior of the unknown function with its derivatives.  Also, a generalization bound is derived to represent the need for having more data points in the steep regions.**

**Rating:** 7
**Confidence:** 4

**Review:**

This paper is well written and very well structured. I think this paper can be interesting for ICLR community. The authors presented experimental results that show marginal improvement using VBSW in both regression and classification applications. The experimental improvement is limited; however, it makes sense since the proposed approach is a preprocessing stage that can not be dynamically tuned during the learning process.
I am interested to see experiments (for synthetic and real datasets) wherein the steep region is noisy, demonstrating how well the proposed strategy works in the presence of noise.
One can expect that focusing too much on the noisy steep regions can be harmful to the learning process.

In my opinion, the investigated problem has been described and motivated properly. It is interesting to formally show (via the presented generalization bound) why NN needs more points in the steep regions and how the steep regions can be formally approximated. The study has proposed reasonable approximations for the cases where the derivations are not available and computing new labels. The theory that backs the statements made in this study seems correct. However, I am concerned about their experimental results that do not seem significant (gain per model is limited). Also, it would be interesting to see a noise robustness investigation in the steep regions.

---

> ### Author Response · Authors · 2020-11-18
> **Response to AnonReviewer2**
>
> Thank you for your positive review. We appreciate that you emphasize the interest of theoretical and formal background that motivates the method, and the derivations that allow making it applicable to realistic ML problems.
>
> Like you said, the error improvement is not as significant as well known breakthroughs like Dropout or Batch normalization. But following the remarks of R2 and R3, we conducted additional experiments that we present in a new section 5.5 called "Complementarity of VBSW", where we show that VBSW is competitive with another recent sample weighting technique, Active Bias [1]. The main aim of this new section is to demonstrate the complementarity of VBSW with other techniques, it also highlights that for the training of a CNN on Cifar10, we obtain a test accuracy of 76.57, 74.94 + 0.10 for VBSW and 76.33, 75.14 + 0.09 for Active bias, with best, mean + se notation.
>
> R1 and R2 comments also emphasized the utility of experiments about VBSW robustness to label noise. Therefore, we decided to add another section (5.4) called "robustness of VBSW". We discuss the robustness of VBSW to weight estimation, and to label noise for a ResNet20 on Cifar10 with different levels of label noise (10%, 20%, 30% and 40%). We find that despite label noise, VBSW still improves the accuracy of the NN for every noise level by 0.33 / 0.20 + 0.01, 0.28 / 0.14 + 0.01, 0.30 / 0.13 + 0.02, 0.69 / 0.53 + 0.02
> respectively. The results is the accuracy gain in %, and presented as best, mean + se. This experimentally shows that VBSW is adapted to a realistic ML task.
>
> We hope that you will appreciate these additional results as well as the updated version of the paper.
>
> [1] Haw-Shiuan Chang, Erik Learned-Miller, and Andrew McCallum. Active bias: Training more accurate neural networks by emphasizing high variance samples. Advances in NeuralInformation Processing Systems 30, 2017

---

### Official Review · AnonReviewer4 · 2020-10-26
**Nice idea, but no experimental comparisons to other algorithms**

**Rating:** 3
**Confidence:** 4

**Review:**

**Summary of paper**

The authors introduce an algorithm called VBSW to re-weight a training data set in order to improve generalization. In summary, VBSW sets the weight of each example to be the sample variance of the labels of its k nearest neighbors. The nearest neighbors are chosen in the embedding space from the second-to-last layer of a pre-trained neural network. The last layer of the pre-trained model is then trained with these new weights.

This approach is quite simple in practice and seems to be theoretically justified.

The authors demonstrate that VBSW achieves better test accuracy than not using VBSW on 3 toy datasets and 5 real-world datasets.

**Conclusions**

Quality: The authors did not experimentally compare VBSW to any alternative algorithms. I find this omission inexcusable. The problem of re-weighting examples to achieve better accuracy has been studied for decades; there are many other algorithms to compare against.

Clarity: The paper is generally well-structured and well-written, although with a few typos and grammatical errors.

Originality: I am not familiar enough with the related work to say whether this idea is novel. However, seems quite simple and potentially very similar to existing published techniques.

**Comments**

Section 3.1 seems to assume that the labels have no noise. For example, if two examples have the same input features, their labels seem to be generated by the same function f, which would always produce the same label for both examples. This seems unrealistic.

Section 2 describes the author's VBSW algorithm as being applied "prior to the training", but the algorithm actually requires a neural network to be pre-trained before the reweighting procedure. I felt the beginning of the paper was misleading in this regard.

**Minor comments**

Table 1: The difference between the mean accuracies of VBSW vs. basline on the BC dataset seems statistically insignificant; I believe VBSW should not be bolded.

---

> ### Author Response · Authors · 2020-11-18
> **Response to AnonReviewer4 (1/2)**
>
> Thank you for your constructive review. We are glad that you appreciate the idea. Your critics allowed us to strengthen the paper.
> As far as we understand, these critics may be summarized in four points:
>
> * **VBSW should be compared with other sample weighting algorithms.**
>
> You share this concern with R2, which motivates us to take advantage of the additional page allowed for the rebuttal revision to add further experiments. Yet, there are two reasons why we initially did not compare VBSW to other Sample Weighting techniques:
>
>
> 1. Sample Weighting algorithms very often aim at solving a specific problem, like class imbalance or noisy labels. Hence, it would be inappropriate to compare them with VBSW in such cases. Actually, the validation benchmarks used in these works are not the same as ours, for these reasons. For instance, [1] uses long tailed data sets for class imbalance and [2] adds noise to the data set for their main validation section. (Note that in our rebuttal revision, in response to R1, R3 and R4, we add a new section called "Robustness of VBSW" where we study the robustness of VBSW to label noise. This section aims at assessing the robustness of VBSW, as a complementary analysis, and not to demonstrate its capacity to solve noisy labels problem. The previous discussion still holds. We make this point clear in the new section.)
> 2. There are still some sample weighting techniques that aim at improving NN's performances in any scenario, like VBSW. However, due to the application of VBSW on the feature space of a NN, they could be used jointly with VBSW.
>
> That being said, since you and R2 both mentioned this point as problematic, we took it into account for our rebuttal revision. We therefore include a new Section (5.5) called "Complementarity of VBSW". We compare VBSW with a general and recent sample weighting technique, Active Bias [3], for the training of NNS on Cifar10 and show that **(1)** VBSW is competitive with Active Bias (76.57 / 74.94 + 0.10 for VBSW and 76.33 / 75.14 + 0.09 for Active bias, with best, mean + se notation) and **(2)** The best results are obtained when we combine VBSW with Active Bias (76.60 / 75.33 + 0.09). This illustrates our point that VBSW can be used jointly with other techniques. We agree that it needed empirical evidence and we thank you for emphasizing it.
>
>
> * **VBSW seems similar to other techniques**
>
>  VBSW is quite simple in its final form, yet it comes from an original idea whose exploitation at the benefit of the training is a theoretical challenge. The derivation of a simple, tractable, but the theoretically justified methodology was an essential part of our research efforts and to our mind, a significant contribution carried by this work.  Also, VBSW may not be the only way of using the initial idea. Like discussed in the discussion, there are ML applications where we have access to the derivatives or the data generating procedure, so we could use it without the derivations that led to VBSW. Appendix A shows an example of such an application.
>
> * **We assume that the data is not noisy (see Section 3.1).**
>
> Section 3.1 aims at introducing the intuition behind VBSW, so the ML task is formalized as a tractable approximation problem to be able to illustrate this intuition quickly. Theoretical results with the addition of label noise could be a perspective for future theoretical work. In our case, the validation section is all the more crucial since it allows testing the method for real-world data sets and so realistic ML tasks.
>
> We understand the need to always empirically test the method in diverse situations. That is why we decided to add another section (5.4) to the main document called "Robustness to VBSW", motivated by this remark and those of R1 and R4. We discuss the robustness of VBSW to weight estimation, and to label noise for a ResNet20 on Cifar10 with different levels of label noise (10%, 20%, 30% and 40%). We find that despite label noise, VBSW still improves the accuracy of the NN for every noise level by 0.33 / 0.20 + 0.01, 0.28 / 0.14 + 0.01, 0.30 / 0.13 + 0.02, 0.69 / 0.53 + 0.02 respectively. The results is the accuracy gain in %, and presented as best / mean + se. This experimentally shows that VBSW is adapted to a realistic ML task.

---

> ### Author Response · Authors · 2020-11-18
> **Response to AnonReviewer4 (2/2)**
>
> * **Presenting VBSW as applied prior to the training is misleading since it requires a pre-trained NN.**
>
> If we are given a dataset with training points and labels, VBSW can theoretically be applied before the training. However, in this work, we indeed focused on the specific case of DL, where we leverage the representation power of NN and apply VBSW in the feature space of the NN, which requires the NN to be pre-trained. This approach allows using VBSW to ML problems where input data is very raw, irregular, and where VBSW, as is, would have been intractable (like discussed in part 4.3). We recognize that we could be made more apparent that the paper focuses on DL, and modify the introduction and some discussion for the rebuttal revision. We also propose to change the title into "Variance Based Sample Weighting for Supervised Deep Learning".
>
>
> We hope that you will find those answers satisfactory and all those modifications and additional content worthy of acceptance.
>
> [1] Yin Cui, Menglin Jia, Tsung-Yi Lin, Yang Song, and Serge Belongie. Class-balanced loss based on effective number of samples. InThe IEEE Conference on Computer Vision and Pattern Recogni-tion (CVPR), June 2019
>
> [2] Tongliang Liu and Dacheng Tao. Classification with noisy labels by importance reweighting. IEEETrans. Pattern Anal. Mach. Intell., 38(3):447–461, March 2016
>
> [3] Haw-Shiuan Chang, Erik Learned-Miller, and Andrew McCallum. Active bias: Training more accurate neural networks by emphasizing high variance sample. Advances in NeuralInformation Processing Systems 30, 2017

---

### Official Review · AnonReviewer3 · 2020-10-26
**Variance based sample weighting**

**Rating:** 6
**Confidence:** 3

**Review:**

A method for computing sample learning weights based on variance is proposed. The method is model independent and a simple k-NN based estimator for the weights is derived. The authors justify their work by appealing to a novel generalisation bound. Overall the idea is interesting but the exposition needs to be significantly improved as proofs are difficult to follow as it currently stands.

# positives

- The idea is interesting and intuitive: essentially concentrate more weight on points that are ambiguous and less on points that are easily learnt
- The approach is potentially quite general covering a large range of learning tasks, though this needs to be carefully evaluated

# cons

- The exposition is very unclear, there are numerous typos and lack of detail in key sections of the proofs
- Though the authors make statements that the method is very general, the empirical evaluation does not validate this. There is no attempt to apply their method to anything other than a NN. Furthermore, many tasks cited by the authors (e.g., class imbalance & noisy labels in the intro) are implied to be covered by this framework, yet are not evaluated. It's difficult to see how this weighting strategy will improve the noisy label scenario in particular.
- The benchmark in experiments is an unweighted NN. While this is a good benchmark, it would be good to see some alternative sample weighting methods since according to the authors "Sample weighting has already been explored in many works and for various goals."

# some examples of confusing sections

- figure 1 lacks axis labels, and panel labels. It takes far too long to read as a result
- page 16: each line states (Jx(theta) <= ...) even though equality holds between the steps. This makes it difficult to see where bounds have been used. Most of this section is simple algebra and could be shortened.
- page 17: The section invoking Hornik's theorem is unclear and need to be expanded with more detail.

# update

The authors have addressed a number of issues and strengthened their submission.

---

> ### Author Response · Authors · 2020-11-18
> **Response to AnonReviewer3**
>
> Thank you for your constructive review. Your critics allowed us to take a step back and will help up significantly improving the paper. We go through the cons point by point.
>
> * **Unclear exposition**
>
> We took the time to proofread the paper again, in order to spot the typos and unclear parts you talk about. We improved the clarity of the proofs, especially the part about Hornik's theorem.
>
>
> * **Generality of VBSW**
>
> Since VBSW is theoretically applicable to any loss based ML models, we presented it as such. We agree that in practice, we focus on VBSW's application to Deep Learning. We are modifying the introduction and some discussions in the paper to clarify that point. Thank you for this comment that helps us make this part more straightforward. We also propose to modify the title of the submission into "Variance Based Sample Weighting for Supervised Deep Learning".
>
> We realize that a sentence was misleading regarding the position of VBSW with respect to other sample weighting techniques that address noisy labels or class imbalance tasks "Its effect is to boost the performances of a NN, whatever the application is". It may have led you to understand that VBSW covers those problems. They are not, and what we meant is that while Sample Weighting algorithms often aim at improving a model's performances by solving a specific issue, like label class imbalance or noisy labels, VBSW is based on a more general idea, applicable in any scenario, even without noise or class imbalance. Therefore, it would be inappropriate to compare VBSW with other mentioned techniques in such tasks, because they were tailored to address these problems while VBSW was not.
>
>
> * **Comparison with other methods**
>
>  We initially did not compare VBSW to other Sample Weighting techniques for two reasons:
>
>
> 1. Like discussed in the previous point, Sample Weighting techniques very often aim at solving a specific problem, like class imbalance or noisy labels. Hence, it would be inappropriate to compare them with VBSW in such cases. The validation benchmarks used in these works are even not the same as ours, for these reasons. For instance, [1] uses long-tailed data sets for class imbalance and [2] adds noise to the data set for their main validation section. (Note that in our rebuttal revision, in response to R1, R3 and R4, we add a new section called "Robustness of VBSW" where we study the robustness of VBSW to label noise. This section aims at assessing the robustness of VBSW, as a complementary analysis, and not to demonstrate its capacity to solve noisy labels problem. The previous discussion still holds, and we make this point clear in the new section.)
> 2. There are still some sample weighting techniques that aim at improving NN's performances in any scenario, like VBSW. However, due to the application of VBSW on the feature space of a NN, they could be used jointly with VBSW.
>
> That being said, you and R3 both mentioned this point as problematic, and we took it into account for our rebuttal revision. We, therefore, include a new Section (5.5) called "Complementarity of VBSW". We compare VBSW with a general and recent sample weighting technique, Active Bias [3], for the training of NNS on Cifar10 and show that **(1)** VBSW is competitive with Active Bias (76.57 / 74.94 + 0.10 for VBSW and 76.33 / 75.14 + 0.09 for Active bias, with best / mean + se notation) and **(2)** The best results are obtained when we combine VBSW with Active Bias (76.60 / 75.33 + 0.09). This behavior illustrates our point that VBSW can be used jointly with other techniques. We agree that it needed empirical evidence and we thank you for emphasizing it.
>
> We are also adding content related to other reviews that might interest you. We describe all the modifications in detail in a general comment. We hope that you will find all these additional sections worthy of acceptance, and that you will be satisfied with our explanations.
>
> [1] Yin Cui, Menglin Jia, Tsung-Yi Lin, Yang Song, and Serge Belongie. Class-balanced loss based on effective number of samples. InThe IEEE Conference on Computer Vision and Pattern Recogni-tion (CVPR), June 2019
>
> [2] Tongliang Liu and Dacheng Tao. Classification with noisy labels by importance reweighting. IEEETrans. Pattern Anal. Mach. Intell., 38(3):447–461, March 2016
>
> [3] Haw-Shiuan Chang, Erik Learned-Miller, and Andrew McCallum. Active bias: Training more accurate neural networks by emphasizing high variance samples. Advances in NeuralInformation Processing Systems 30, 2017

---

### Official Review · AnonReviewer1 · 2020-10-29
**Intuitive but nontrivial method supported by experiments**

**Rating:** 6
**Confidence:** 2

**Review:**

The manuscript follows an intuitively straightforward conclusion: data should focus on the region where the function to learning is steeper. Based on that, a non-trivial method is proposed to improve the performance of NN. Empirical results are provided to support the performance of the proposed method.

According to my understanding, the performance gain from the proposed sampling method mainly comes from the sample complexity part, or in other words, the variance part. Is this correct? The weight approximation method proposed method relies on the derivative of the estimated NN. Will this cause much randomness or error to the approximated weights, since making sure the derivative of the estimation is accurate is more challenging than making the estimation itself is accurate?

Since provided analysis is mostly on the asymptotic level, it is quite unclear that how much noise/error the proposed sampling method may have, how much these noise will hurt the final NN estimation. So, is there any cases where the proposed sampling method will even hurt the performance of the NN? (For example, what will happen if the weights are poorly approximated?) If not, what are the key conditions to make sure that the sampling method really provides significant performance gains?

Also, can we iteratively implement the weight approximation and sampling method many times in Algorithm 1? If the proposed sampling method really helps the performance, we can have more accurate weight approximations, and further improve the performance....

In all, the method is interesting with improved performance supported by experiments. However, I believe more analysis on the constraints and potential error of the method will benefit the manuscript.

---

> ### Author Response · Authors · 2020-11-18
> **Response to AnonReviewer1**
>
> Thank you very much for your positive review. We identified four main questions in your review:
>
> ### Questions
>
> * **Q1: Does VBSW performance gain come from the variance part ?**
> You are right to say that the performance gain comes from the variance part. Indeed, the value of the local variance measures the local complexity of the function f to learn, so makes the model focus its learning efforts on the regions where f is more challenging to learn.
> * **Q2: Will working with the derivative bring more noise, since making both the derivatives of the estimation and the estimation itself accurate is a more challenging task ?**
> The approximation method does not rely on the derivatives of the estimated NN but of the function f to learn. The classical estimation or approximation of f with a NN and the estimation of f's derivatives are two different problems. Yet, the estimation of the derivatives indeed is a challenging task. Experiments of Section 5 aims at evaluating if the estimation error and noise introduced in Section 4.1, still allows the method to work. Results show that these approximations are reasonable.
> * **Q3: To what extent the noise brought by the estimation can hurt the performance, and are there any scenarios where VBSW does not improve the accuracy ?**
> Like it is studied in Appendix C, many different VBSW's hyperparameters combinations can lead to performance improvement, which suggests that VBSW is robust to noise in the estimation of the weights. We include a new section in the rebuttal revision which studies the robustness of VBSW, where we discuss this point.
> VBSW may be limited in cases where the input or output dimension is too high.
> The input dimension is discussed in Section 4.3. In that case, the limitations come from the KNN part. Most KNN algorithms suffer from the curse of dimensionality. Besides, for high dimensional spaces like image, the structure of data makes the concept of nearest neighbors irrelevant. In this paper, it is alleviated by applying VBSW in the lower dimensional feature space of NNs.
> The output dimension can be explained by the fact that variance does not have any sense for too highly dimensional and complex data. This behavior could happen, for instance, when the NN would have to generate images like in video prediction.
> * **Q4: Could we apply VBSW iteratively to obtain a better estimation of the weights ?**
> Even if the mathematical background of VBSW relies on statistical estimation and probability, the final method is deterministic since it involves KNN on the input space to compute the variance. So Iteratively applying VBSW would not change the estimated value of the weights.
>
> ### Other comments
>
> We agree with the concerns presented about the estimation error of the weights, analysis of the constraints and limits of the method. That is why we include a new Section (5.4) in the rebuttal revision, called "Robustness of VBSW", where we add discussions about the robustness of VBSW to estimation error. We explain that VBSW weights estimation is robust to error since, as suggests the hyperparameters study in section 5.1, many different values of the weights lead to performance improvement.
>
>  We include experiments to test robustness of VBSW to label noise for a ResNet20 on Cifar10 with different levels of label noise (10%, 20%, 30% and 40%). We find that despite label noise, VBSW still improves the accuracy of the NN for every noise level by 0.33 / 0.20 + 0.01, 0.28 / 0.14 + 0.01, 0.30 / 0.13 + 0.02, 0.69 / 0.53 + 0.02 respectively. The results is the accuracy gain in %, and presented as best, mean + se.
>
> We also add content related to other reviews that might interest you. We describe all the modifications in detail in the general comment. We hope that you will appreciate these additional sections.

---

### Author Response · Authors · 2020-11-18
**Detailed modifications of the paper**

In addition to many typos corrections and sentence reformulations, we updated the pdf and made the following modifications:
* Title updated to "Variance Based Sample Weighting for Supervised Deep Learning".
* page 1 - second paragraph of the introduction: amendment to remove “prior to the training and regardless of the model”
* page 1, 2 - last paragraph of the introduction: modifications to clarify that the paper focuses on Deep Learning
* page 2 - last paragraph of the introduction: modification to introduce the new experiments.
* page 2 - at the end of Active Learning paragraph: modification to remove ‘before the training”
* page 2 - at the end of Example weighting paragraph: modifications to make it more clear that VBSW does not claim to solve any data set specific problems like class imbalance or noisy labels.
* page 3 - Figure 1: modifications to include axis labels and panel labels.
* page 6 - introduction of section 5: modification to introduce the new subsections 5.4: Robustness of VBSW and 5.5: Complementarity of VBSW.
* page 8 - Added new sections 5.4: Robustness of VBSW and 5.5: Complementarity of VBSW that include **(1)** experiments to study the robustness of VBSW to noise for a ResNet on noisy Cifar10 (5.4) and the complementarity VBSW for DL with other weighting techniques on Cifar10 (5.5),  And **(2)** discussions about the robustness of VBSW to noise estimation error (5.4) and the efficiency of VBSW (5.5)
* page 9 - last paragraph of discussion: modification to refer to Appendix A for examples of ML applications exploiting the motivating idea (based on derivatives of the function to learn) without using VBSW.
* page 9 - modification of the conclusion to make it more clear that this work focuses on Deep Learning.
* page 17 - Concatenation of derivations
* page 18 - Clarification of Hornik's theorem statement and utilization to obtain result (9)


**Update 20/11**

We have made further typos corrections on the paper.

---

### Author Response · Authors · 2020-11-18
**General comment**

We want to thank the reviewers for their constructive reviews.
Here, we first provide a summary of the main points of the review. We then comment these points in a second part. These comments are detailed in responses to reviewers.

### Main points raised by the reviewers:

1. VBSW comes from an interesting idea that a Neural Network needs to focus more  on regions where the function to learn is steeper. This idea is theoretically illustrated by a generalization bound, which makes it well motivated. The performance improvement is supported by experiments.
2. Since VBSW relies on statistical estimation, the paper would benefit from further analyses on the robustness of VBSW to label noise or weight estimation error.
3. VBSW is a sample weighting algorithm, yet it is not compared to other sample weighting algorithms.
4. Although VBSW is general and applicable to any Machine Learning model, the paper’s experiments only focus on Neural Networks.

### Author’s comments

1. Constructing a practical and straightforward method from this original idea was a theoretical challenge, which was an essential part of our research efforts. To our mind, it is a critical contribution carried by this work. Moreover, the idea itself could be used for other applications than sample weighting. Appendix A introduces an example of such an application, that uses the derivatives of $f$ to sample more efficient training data points.
2. We thank you for this remark that allowed us to conduct additional analyses and reflections on the robustness of VBSW. This work led to a new Section (5.4) called “Robustness of VBSW”, where we study VBSW performance improvement in the presence of label noise. We trained a ResNet20 on Cifar10 with different levels of label noise (10%, 20%, 30% and 40%). We find that despite label noise, VBSW still improves the accuracy of the NN for every noise level by 0.33 / 0.20 + 0.01, 0.28 / 0.14 + 0.01, 0.30 / 0.13 + 0.02 and 0.69 / 0.53 + 0.02 respectively. Note that this accuracy gain is in %, and presented as best / mean + se. Please see the updated pdf for more details.
3. The absence of comparison with other Sample Weighting techniques was initially not an omission to us. Indeed, most sample weighting techniques are tailored to address specific problems, like label noise or class imbalance, whereas VBSW comes from an idea about learning theory. Moreover, its application to Deep Learning, i.e. in the feature space of NNs, makes it applicable jointly with any other weighting techniques. However, the reviews made us realize that this statement needed empirical evidence. Hence, we include a new Section (5.5) called "Complementarity of VBSW". We compare VBSW with a general and recent sample weighting technique, Active Bias [3], for the training of NNS on Cifar10 and show that
VBSW is competitive with Active Bias (76.57 / 74.94 + 0.10 for VBSW and 76.33 / 75.14 + 0.09 for Active bias, with best / mean + se notation) and
The best results are obtained when we combine VBSW with Active Bias (76.60 / 75.33 + 0.09), which illustrates our point that VBSW can be used jointly with other techniques.
    More details can be found in the updated pdf.
4. It is true that theoretically, VBSW could be applied to any loss function based models. We chose to focus our working efforts on the specific case of Deep Learning, which brings an exciting reflection around the application on the feature space. Therefore, we rephrased some sentences in the introduction and discussions to make this point more straightforward. We also propose to change the title of the paper into “ Variance Based Sample Weighting for Supervised Deep Learning”.

Full details about the paper modifications are described in a dedicated comment.

---

### Decision · Program_Chairs · 2021-01-07
**Final Decision**

**Decision:**

Reject

**Comment:**

# Paper Summary

This paper proposes "variance based sample weighting" (VBSW). The key observation is that, in areas where the labeling function is changing rapidly, more samples may be required to achieve a good fit. In Section 3, they justify this intuition by showing that areas in which the label gradient is higher disproportionately impact the generalization performance, and go on to propose upweighting examples proportionately to the local label variance.

This label variance is approximated, for each training example, by finding its k-nearest-neighbors (either in the feature space, or a latent space--they use the latter in their experiments), and calculating the sample variance of the labels. There's a bit of a leap here: the authors started advocating for drawing more samples from rapidly-changing regions, but ended up upweighting the existing samples. These are not the same thing, but this issue is not discussed.

The experiments are well-thought out and comprehensive (although Reviewer 4 complained about a lack of nontrivial baselines, and I agree). The first and fourth set of experiments are particularly impressive. The first uses toy datasets to enable easy visualization, while the fourth (added during the response based on a comment of Reviewer 1) explores how robust VBSW is to hyperparameter choices and label noise. The second and third sets of experiments are on "realistic" problems, and while the gains are arguably marginal, VBSW does show consistently positive results.

# Pros

1. The reviewers agreed that the fundamental idea was intuitive and well-explained
1. The algorithm is general, and easy to implement
1. Reviewer 1 asked how robust VBSW was to hyperparameter choices and label noise, and the authors added a new section that I think does an excellent job alleviating such concerns
1. Experiments are comprehensive, including both toy datasets on which the behavior of the algorithm can be easily visualized, and more realistic experiments on MNIST, CIFAR 10,  RTE, STS-B AND MRPC. They show consistently positive results (although not especially large ones)

# Cons

1. The reviewers had mixed opinions on the writing quality, although they generally agreed that it was well-organized. There are a large number of typos, grammatical errors, and awkward phrases
1. Reviewer 4 noted that Section 3 assumes that there is no label noise, which seems unrealistic
1. Reviewers 2 and 4 complained about a lack of nontrivial baselines in the experiments (reviewer 4 called this "inexcusable"). In response, the authors added a new experiment comparing with active bias, on which the results were arguably positive, but not convincingly so. Regardless, this is only one experiment--there should be baselines for all experiments (except perhaps for the ones on UCI datasets), ideally multiple baselines (would a curriculum learning approach be appropriate on any of these tasks?)

# Conclusion

Three of the four reviewers recommended acceptance, but reviewer 4 gave a very negative score (3: clear rejection). Most of this reviewer's criticisms were fairly minor, with the two major ones being (i) that Section 3 unrealistically assumes that there is no label noise, and (ii) that the experiments have no nontrivial baselines. This first criticism is, I think, not a *huge* deal: Section 3 is only intended to provide intuition. The authors attempted to address the second criticism by adding one experiment comparing to "active bias", but I think that this is insufficient. In addition, while the paper is well-structured, I agree with the reviewers who complained about the paper's lack of polish.

All reviewers praised the fundamental idea, and said that the authors gave good intuition for their approach. The experimental results are also fairly comprehensive (aside from the lack of nontrivial baselines), and show positive results. The new section on robustness (in response to Reviewer 1) is a great addition that, I think, fills in most of the remaining "gaps" in this work. The major outstanding issues, in my opinion, are the writing quality, and the lack of nontrivial experimental baselines. These are very fixable, but I think that they're too significant for the paper to be accepted. Overall, I think that this is a borderline paper, but it's on the rejection side of the boundary.